# Finding Counterfactually Optimal Action Sequences in Continuous State Spaces

**Stratis Tsirtsis**
Max Planck Institute for Software Systems
Kaiserslautern, Germany
stsirtsis@mpi-sws.org

**Manuel Gomez-Rodriguez**
Max Planck Institute for Software Systems
Kaiserslautern, Germany
manuelgr@mpi-sws.org

## Abstract

Whenever a clinician reflects on the efficacy of a sequence of treatment decisions for a patient, they may try to identify critical time steps where, had they made different decisions, the patient's health would have improved. While recent methods at the intersection of causal inference and reinforcement learning promise to aid human experts, as the clinician above, to *retrospectively* analyze sequential decision making processes, they have focused on environments with finitely many discrete states. However, in many practical applications, the state of the environment is inherently continuous in nature. In this paper, we aim to fill this gap. We start by formally characterizing a sequence of discrete actions and continuous states using finite horizon Markov decision processes and a broad class of bijective structural causal models. Building upon this characterization, we formalize the problem of finding counterfactually optimal action sequences and show that, in general, we cannot expect to solve it in polynomial time. Then, we develop a search method based on the $A^*$ algorithm that, under a natural form of Lipschitz continuity of the environment's dynamics, is guaranteed to return the optimal solution to the problem. Experiments on real clinical data show that our method is very efficient in practice, and it has the potential to offer interesting insights for sequential decision making tasks.

## 1 Introduction

Had the chess player moved the king one round later, would they have avoided losing the game? Had the physician administered antibiotics one day earlier, would the patient have recovered? The process of mentally simulating alternative worlds where events of the past play out differently than they did in reality is known as counterfactual reasoning [1]. Thoughts of this type are a common by-product of human decisions and they are tightly connected to the way we attribute causality and responsibility to events and others' actions [2]. The last decade has seen a rapid development of reinforcement learning agents, presenting (close to) human-level performance in a variety of sequential decision making tasks, such as gaming [3, 4], autonomous driving [5] and clinical decision support [6, 7]. In conjunction with the substantial progress made in the field of causal inference [8, 9], this has led to a growing interest in machine learning methods that employ elements of counterfactual reasoning to improve or to retrospectively analyze decisions in sequential settings [10–15].

In the context of reinforcement learning, sequential decision making is typically modeled using Markov Decision Processes (MDPs) [16]. Here, we consider MDPs with a finite horizon where each episode (*i.e.*, each sequence of decisions) consists of a finite number of time steps. As an example, consider a clinician treating a patient in an intensive care unit (ICU). At each time step, the clinician observes the current state of the environment (*e.g.*, the patient's vital signs) and they choose among a set of potential actions (*e.g.*, standardized dosages of a drug). Consequently, the chosen action causes the environment to transition (stochastically) into a new state, and the clinician earns a reward (*e.g.*,

37th Conference on Neural Information Processing Systems (NeurIPS 2023).

satisfaction inversely proportional to the patient's severity). The process repeats until the horizon is met and the goal of the clinician is to maximize the total reward.

In this work, our goal is to aid the retrospective analysis of individual episodes as the example above. For each episode, we aim to find an action sequence that differs slightly from the one taken in reality but, under the circumstances of that particular episode, would have led to a higher counterfactual reward. In our example above, assume that the patient's condition does not improve after a certain period of time. A counterfactually optimal action sequence could highlight to the clinician a small set of time steps in the treatment process where, had they administered different drug dosages, the patient's severity would have been lower. In turn, a manual inspection of those time steps could provide insights to the clinician about potential ways to improve their treatment policy.

To infer how a particular episode would have evolved under a different action sequence than the one taken in reality, one needs to represent the stochastic state transitions of the environment using a structural causal model (SCM) [8, 17]. This has been a key aspect of a line of work at the intersection of counterfactual reasoning and reinforcement learning, which has focused on methods to either design better policies using offline data [10, 12] or to retrospectively analyze individual episodes [11, 13]. Therein, the work most closely related to ours is by Tsirtsis et al. [13], which introduces a method to compute counterfactually optimal action sequences in MDPs with discrete states and actions using a Gumbel-Max SCM to model the environment dynamics [11]. However, in many practical applications, such as in critical care, the state of the environment is inherently continuous in nature [18]. In our work, we aim to fill this gap by designing a method to compute counterfactually optimal action sequences in MDPs with continuous states and discrete actions. Refer to Appendix A for a discussion of further related work and to Pearl [8] for an overview of the broad field of causality.

**Our contributions.** We start by formally characterizing sequential decision making processes with continuous states and discrete actions using finite horizon MDPs and a general class of *bijective* SCMs [19]. Notably, this class of SCMs includes multiple models introduced in the causal discovery literature [20–25]. Building on this characterization, we make the following contributions:

  (i) We formalize the problem of finding a counterfactually optimal action sequence for a particular episode in environments with continuous states under the constraint that it differs from the observed action sequence in at most $k$ actions.
 (ii) We show that the above problem is NP-hard using a novel reduction from the classic partition problem [26]. This is in contrast with the computational complexity of the problem in environments with discrete states, which allows for polynomial time algorithms [13].
(iii) We develop a search method based on the $A^*$ algorithm that, under a natural form of Lipschitz continuity of the environment's dynamics, is guaranteed to return the optimal solution to the problem upon termination.

Finally, we evaluate the performance and the qualitative insights of our method by performing a series of experiments using real patient data from critical care.[1]

## 2  A causal model of sequential decision making processes

At each time step $t \in [T-1] := \{0, 1, \ldots, T-1\}$, where $T$ is a time horizon, the decision making process is characterized by a $D$-dimensional vector state $\boldsymbol{s}_t \in \mathcal{S} = \mathbb{R}^D$, an action $a_t \in \mathcal{A}$, where $\mathcal{A}$ is a finite set of $N$ actions, and a reward $R(\boldsymbol{s}_t, a_t) \in \mathbb{R}$ associated with each pair of states and actions. Moreover, given an episode of the decision making process, $\tau = \{(\boldsymbol{s}_t, a_t)\}_{t=0}^{T-1}$, the process's outcome $o(\tau) = \sum_t R(\boldsymbol{s}_t, a_t)$ is given by the sum of the rewards. In the remainder, we will denote the elements of a vector $\boldsymbol{s}_t$ as $s_{t,1}, \ldots, s_{t,D}$.[2]

Further, we characterize the dynamics of the decision making process using the framework of structural causal models (SCMs). In general, an SCM is consisted of four parts: (i) a set of endogenous variables (ii) a set of exogenous (noise) variables (iii) a set of structural equations assigning values to the endogenous variables, and (iv) a set of prior distributions characterizing the exogenous variables [8]. In our setting, the endogenous variables of the SCM $\mathcal{C}$ are the random variables representing the states $\boldsymbol{S}_0, \ldots, \boldsymbol{S}_{T-1}$ and the actions $A_0, \ldots, A_{T-1}$. The action $A_t$ at time

---

[1]Our code is accessible at https://github.com/Networks-Learning/counterfactual-continuous-mdp.
[2]Table 1 in Appendix B summarizes the notation used throughout the paper.

step $t$ is chosen based on the observed state $S_t$ and is given by a structural (policy) equation

$$A_t := g_A(S_t, Z_t), \tag{1}$$

where $Z_t \in \mathcal{Z}$ is a vector-valued noise variable, to allow some level of stochasticity in the choice of the action, and its prior distribution $P^{\mathcal{C}}(Z_t)$ is characterized by a density function $f^{\mathcal{C}}_{Z_t}$. Similarly, the state $S_{t+1}$ in the next time step is given by a structural (transition) equation

$$S_{t+1} := g_S(S_t, A_t, U_t), \tag{2}$$

where $U_t \in \mathcal{U}$ is a vector-valued noise variable with its prior distribution $P^{\mathcal{C}}(U_t)$ having a density function $f^{\mathcal{C}}_{U_t}$, and we refer to the function $g_S$ as the *transition mechanism*. Note that, in Eq. 2, the noise variables $\{U_t\}_{t=0}^{T-1}$ are mutually independent and, keeping the sequence of actions fixed, they are the only source of stochasticity in the dynamics of the environment. In other words, a sampled sequence of noise values $\{u_t\}_{t=0}^{T-1}$ and a fixed sequence of actions $\{a_t\}_{t=0}^{T-1}$ result into a single (deterministic) sequence of states $\{s_t\}_{t=0}^{T-1}$. This implicitly assumes that the state transitions are stationary and there are no unobserved confounders. Figure 4 in Appendix B depicts the causal graph $G$ corresponding to the SCM $\mathcal{C}$ defined above.

The above representation of sequential decision making using an SCM $\mathcal{C}$ is a more general reformulation of a Markov decision process, where a (stochastic) policy $\pi(a \mid s)$ is entailed by Eq. 1, and the transition distribution (*i.e.*, the conditional distribution of $S_{t+1} \mid S_t, A_t$) is entailed by Eq. 2. Specifically, the conditional density function of $S_{t+1} \mid S_t, A_t$ is given by

$$p^{\mathcal{C}}(S_{t+1} = s \mid S_t = s_t, A_t = a_t) = p^{\mathcal{C}\,;\,do(A_t=a_t)}(S_{t+1} = s \mid S_t = s_t)$$
$$= \int_{u \in \mathcal{U}} \mathbb{1}[s = g_S(s_t, a_t, u)] \cdot f^{\mathcal{C}}_{U_t}(u) du, \quad (3)$$

where $do(A_t = a_t)$ denotes a (hard) intervention on the variable $A_t$, whose value is set to $a_t$.[3] Here, the first equality holds because $S_{t+1}$ and $A_t$ are d-separated by $S_t$ in the sub-graph obtained from $G$ after removing all outgoing edges of $A_t$[4] and the second equality follows from Eq. 2.

Moreover, as argued elsewhere [11, 13], by using an SCM to represent sequential decision making, instead of a classic MDP, we can answer counterfactual questions. More specifically, assume that, at time step $t$, we observed the state $S_t = s_t$, we took action $A_t = a_t$ and the next state was $S_{t+1} = s_{t+1}$. Retrospectively, we would like to know the probability that the state $S_{t+1}$ would have been $s'$ if, at time step $t$, we had been in a state $s$, and we had taken an action $a$, (generally) different from $s_t, a_t$. Using the SCM $\mathcal{C}$, we can characterize this by a counterfactual transition density function

$$p^{\mathcal{C} \mid S_{t+1}=s_{t+1}, S_t=s_t, A_t=a_t\,;\,do(A_t=a)}(S_{t+1} = s' \mid S_t = s) =$$
$$\int_{u \in \mathcal{U}} \mathbb{1}[s' = g_S(s, a, u)] \cdot f^{\mathcal{C} \mid S_{t+1}=s_{t+1}, S_t=s_t, A_t=a_t}_{U_t}(u) du, \quad (4)$$

where $f^{\mathcal{C} \mid S_{t+1}=s_{t+1}, S_t=s_t, A_t=a_t}_{U_t}$ is the posterior distribution of the noise variable $U_t$ with support such that $s_{t+1} = g_S(s_t, a_t, u)$.

In what follows, we will assume that the transition mechanism $g_S$ is continuous with respect to its last argument and the SCM $\mathcal{C}$ satisfies the following form of Lipschitz-continuity:

**Definition 1.** *An SCM $\mathcal{C}$ is Lipschitz-continuous iff the transition mechanism $g_S$ and the reward $R$ are Lipschitz-continuous with respect to their first argument,* i.e., *for each $a \in \mathcal{A}$, $u \in \mathcal{U}$, there exists a Lipschitz constant $K_{a,u} \in \mathbb{R}_+$ such that, for any $s, s' \in \mathcal{S}$, $\|g_S(s, a, u) - g_S(s', a, u)\| \leq K_{a,u} \|s - s'\|$, and, for each $a \in \mathcal{A}$, there exists a Lipschitz constant $C_a \in \mathbb{R}_+$ such that, for any $s, s' \in \mathcal{S}$, $|R(s, a) - R(s', a)| \leq C_a \|s - s'\|$. In both cases, $\|\cdot\|$ denotes the Euclidean distance.*

Note that, although they are not phrased in causal terms, similar Lipschitz continuity assumptions for the environment dynamics are common in prior work analyzing the theoretical guarantees of reinforcement learning algorithms [27–35]. Moreover, for practical applications (*e.g.*, in healthcare), this is a relatively mild assumption to make. Consider two patients whose vitals $s$ and $s'$ are similar

---

[3]In general, the *do* operator also allows for soft interventions (*i.e.*, setting a probability distribution for $A_t$).

[4]This follows directly from the rules of *do*-calculus. For further details, refer to Chapter 3 of Pearl [8].

at a certain point in time, they receive the same treatment $a$, and every unobserved factor $\boldsymbol{u}$ that may affect their health is also the same. Intuitively, Definition 1 implies that their vitals will also evolve similarly in the immediate future, *i.e.*, the values $g_S(\boldsymbol{s}, a, \boldsymbol{u})$ and $g_S(\boldsymbol{s}', a, \boldsymbol{u})$ will not differ dramatically. In this context, it is worth mentioning that, when the transition mechanism $g_S$ is modeled by a neural network, it is possible to control its Lipschitz constant during training, and penalizing high values can be seen as a regularization method [36, 37].

Further, we will focus on bijective SCMs [19], a fairly broad class of SCMs, which subsumes multiple models studied in the causal discovery literature, such as additive noise models [20], post-nonlinear causal models [21], location-scale noise models [22] and more complex models with neural network components [23–25].

**Definition 2.** *An SCM $\mathcal{C}$ is bijective iff the transition mechanism $g_S$ is bijective with respect to its last argument,* i.e.*, there is a well-defined inverse function $g_S^{-1} : \mathcal{S} \times \mathcal{A} \times \mathcal{S} \to \mathcal{U}$ such that, for every combination of $\boldsymbol{s}_{t+1}, \boldsymbol{s}_t, a_t, \boldsymbol{u}_t$ with $\boldsymbol{s}_{t+1} = g_S(\boldsymbol{s}_t, a_t, \boldsymbol{u}_t)$, it holds that $\boldsymbol{u}_t = g_S^{-1}(\boldsymbol{s}_t, a_t, \boldsymbol{s}_{t+1})$.*

Importantly, bijective SCMs allow for a more concise characterization of the counterfactual transition density given in Eq. 4. More specifically, after observing an event $\boldsymbol{S}_{t+1} = \boldsymbol{s}_{t+1}, \boldsymbol{S}_t = \boldsymbol{s}_t, A_t = a_t$, the value $\boldsymbol{u}_t$ of the noise variable $\boldsymbol{U}_t$ can only be such that $\boldsymbol{u}_t = g_S^{-1}(\boldsymbol{s}_t, a_t, \boldsymbol{s}_{t+1})$, *i.e.*, the posterior distribution of $\boldsymbol{U}_t$ is a point mass and its density is given by

$$f_{\boldsymbol{U}_t}^{\mathcal{C} \,|\, \boldsymbol{S}_{t+1}=\boldsymbol{s}_{t+1}, \boldsymbol{S}_t=\boldsymbol{s}_t, A_t=a_t}(\boldsymbol{u}) = \mathbb{1}[\boldsymbol{u} = g_S^{-1}(\boldsymbol{s}_t, a_t, \boldsymbol{s}_{t+1})], \tag{5}$$

where $\mathbb{1}[\cdot]$ denotes the indicator function. Then, for a given episode $\tau$ of the decision making process, we have that the (non-stationary) counterfactual transition density is given by

$$p_{\tau,t}(\boldsymbol{S}_{t+1} = \boldsymbol{s}' \,|\, \boldsymbol{S}_t = \boldsymbol{s}, A_t = a) := p^{\mathcal{C} \,|\, \boldsymbol{S}_{t+1}=\boldsymbol{s}_{t+1}, \boldsymbol{S}_t=\boldsymbol{s}_t, A_t=a_t \,;\, do(A_t=a)}(\boldsymbol{S}_{t+1} = \boldsymbol{s}' \,|\, \boldsymbol{S}_t = \boldsymbol{s})$$

$$= \int_{\boldsymbol{u} \in \mathcal{U}} \mathbb{1}[\boldsymbol{s}' = g_S(\boldsymbol{s}, a, \boldsymbol{u})] \cdot \mathbb{1}[\boldsymbol{u} = g_S^{-1}(\boldsymbol{s}_t, a_t, \boldsymbol{s}_{t+1})] d\boldsymbol{u}$$

$$= \mathbb{1}\left[\boldsymbol{s}' = g_S\left(\boldsymbol{s}, a, g_S^{-1}(\boldsymbol{s}_t, a_t, \boldsymbol{s}_{t+1})\right)\right]. \tag{6}$$

Since this density is also a point mass, the resulting counterfactual dynamics are purely deterministic. That means, under a bijective SCM , the answer to the question "*What would have been the state at time $t + 1$, had we been at state $\boldsymbol{s}$ and taken action $a$ at time $t$, given that, in reality, we were at $\boldsymbol{s}_t$, we took $a_t$ and the environment transitioned to $\boldsymbol{s}_{t+1}$?*" is just given by $\boldsymbol{s}' = g_S\left(\boldsymbol{s}, a, g_S^{-1}(\boldsymbol{s}_t, a_t, \boldsymbol{s}_{t+1})\right)$.

**On the counterfactual identifiability of bijective SCMs.** Very recently, Nasr-Esfahany and Kiciman [38] have shown that bijective SCMs are in general not counterfactually identifiable when the exogenous variable $\boldsymbol{U}_t$ is multi-dimensional. In other words, even with access to an infinite amount of triplets $(\boldsymbol{s}_t, a_t, \boldsymbol{s}_{t+1})$ sampled from the true SCM $\mathcal{C}$, it is always possible to find an SCM $\mathcal{M} \neq \mathcal{C}$ with transition mechanism $h_S$ and distributions $P^{\mathcal{M}}(\boldsymbol{U}_t)$ that entails the same transition distributions as $\mathcal{C}$ (*i.e.*, it fits the observational data perfectly), but leads to different counterfactual predictions. Although our subsequent algorithmic results do not require the SCM $\mathcal{C}$ to be counterfactually identifiable, the subclass of bijective SCMs we will use in our experiments in Section 5 is counterfactually identifiable. The defining attribute of this subclass, which we refer to as *element-wise bijective SCMs*, is that the transition mechanism $g_S$ can be decoupled into $D$ independent mechanisms $g_{S,i}$ such that $S_{t+1,i} = g_{S,i}(\boldsymbol{S}_t, A_t, U_{t,i})$ for $i \in \{1, \ldots, D\}$. This implies $S_{t+1,i} \perp\!\!\!\perp U_{t,j} \,|\, U_{t,i}, \boldsymbol{S}_t, A_t$ for $j \neq i$, however, $U_{t,i}, U_{t,j}$ do not need to be independent. Informally, we have the following identifiability result (refer to Appendix C for a formal version of the theorem along with its proof, which follows a similar reasoning to proofs found in related work [12, 19]):

**Theorem 3** (Informal). *Let $\mathcal{C}$ and $\mathcal{M}$ be two element-wise bijective SCMs such that their entailed transition distributions for $\boldsymbol{S}_{t+1}$ given any value of $\boldsymbol{S}_t, A_t$ are always identical. Then, all their counterfactual predictions based on an observed transition $(\boldsymbol{s}_t, a_t, \boldsymbol{s}_{t+1})$ will also be identical.*

**On the assumption of no unobserved confounding.** The assumption that there are no hidden confounders is a frequent assumption made by work at the intersection of counterfactual reasoning and reinforcement learning [10–13] and, more broadly, in the causal inference literature [39–43]. That said, there is growing interest in developing off-policy methods for partially observable MPDs (POMDPs) that are robust to certain types of confounding [44–46], and in learning dynamic treatment regimes in sequential settings with non-Markovian structure [47, 48]. Moreover, there is a line of work focusing on the identification of counterfactual quantities in non-sequential confounded environments [49–51]. In that context, we consider the computation of (approximately) optimal counterfactual action sequences under confounding as a very interesting direction for future work.

## 3 Problem statement

Let $\tau$ be an observed episode of a decision making process whose dynamics are characterized by a Lipschitz-continuous bijective SCM. To characterize the counterfactual outcome that any alternative action sequence would have achieved under the circumstances of the particular episode, we build upon the formulation of Section 2, and we define a non-stationary counterfactual MDP $\mathcal{M}^+ = (\mathcal{S}^+, \mathcal{A}, F^+_{\tau,t}, R^+, T)$ with deterministic transitions. Here, $\mathcal{S}^+ = \mathcal{S} \times [T-1]$ is an enhanced state space such that each $s^+ \in \mathcal{S}^+$ is a pair $(s, l)$ indicating that the counterfactual episode would have been at state $s \in \mathcal{S}$ with $l$ action changes already performed. Accordingly, $R^+$ is a reward function which takes the form $R^+((s, l), a) = R(s, a)$ for all $(s, l) \in \mathcal{S}^+$, $a \in \mathcal{A}$, i.e., it does not change depending on the number of action changes already performed. Finally, the time-dependent transition function $F^+_{\tau,t} : \mathcal{S}^+ \times \mathcal{A} \to \mathcal{S}^+$ is defined as

$$F^+_{\tau,t}((s, l), a) = \begin{cases} \left(g_S\left(s, a, g_S^{-1}(s_t, a_t, s_{t+1})\right), l+1\right) & \text{if } (a \neq a_t) \\ \left(g_S\left(s, a_t, g_S^{-1}(s_t, a_t, s_{t+1})\right), l\right) & \text{otherwise.} \end{cases} \tag{7}$$

Intuitively, here we set the transition function according to the point mass of the counterfactual transition density given in Eq. 6, and we use the second coordinate to keep track of the changes that have been performed in comparison to the observed action sequence up to the time step $t$.

Now, given the initial state $s_0$ of the episode $\tau$ and any counterfactual action sequence $\{a'_t\}_{t=0}^{T-1}$, we can compute the corresponding counterfactual episode $\tau' = \{(s'_t, l_t), a'_t\}_{t=0}^{T-1}$. Its sequence of states is given recursively by

$$(s'_1, l_1) = F^+_{\tau,0}((s_0, 0), a'_0) \quad \text{and} \quad (s'_{t+1}, l_{t+1}) = F^+_{\tau,0}((s'_t, l_t), a'_t) \text{ for } t \in \{1, \ldots, T-1\}, \tag{8}$$

and its counterfactual outcome is given by $o^+(\tau') := \sum_t R^+((s'_t, l_t), a'_t) = \sum_t R(s'_t, a'_t)$.

Then, similarly as in Tsirtsis et al. [13], our ultimate goal is to find the counterfactual action sequence $\{a'_t\}_{t=0}^{T-1}$ that, starting from the observed initial state $s_0$, maximizes the counterfactual outcome subject to a constraint on the number of counterfactual actions that can differ from the observed ones, i.e.,

$$\underset{a'_0, \ldots, a'_{T-1}}{\text{maximize}} \quad o^+(\tau') \qquad \text{subject to} \quad s'_0 = s_0 \text{ and } \sum_{t=0}^{T-1} \mathbf{1}[a_t \neq a'_t] \leq k, \tag{9}$$

where $a_0, \ldots, a_{T-1}$ are the observed actions. Unfortunately, using a reduction from the classic partition problem [26], the following theorem shows that we cannot hope to find the optimal action sequence in polynomial time:[5]

**Theorem 4.** *The problem defined by Eq. 9. is NP-Hard.*

The proof of the theorem relies on a reduction from the partition problem [26], which is known to be NP-complete, to our problem, defined in Eq. 9. At a high-level, we map any instance of the partition problem to an instance of our problem, taking special care to construct a reward function and an observed action sequence, such that the optimal counterfactual outcome $o^+(\tau^*)$ takes a specific value if and only if there exists a valid partition for the original instance. The hardness result of Theorem 4 motivates our subsequent focus on the design of a method that *always* finds the optimal solution to our problem at the expense of a potentially higher runtime for some problem instances.

## 4 Finding the optimal counterfactual action sequence via A* search

To deal with the increased computational complexity of the problem, we develop an optimal search method based on the classic $A^*$ algorithm [52], which we have found to be very efficient in practice. Our starting point is the observation that, the problem of Eq. 9 presents an optimal substructure, i.e., its optimal solution can be constructed by combining optimal solutions to smaller sub-problems. For an observed episode $\tau$, let $V_\tau(s, l, t)$ be the maximum counterfactual reward that could have been achieved in a counterfactual episode where, at time $t$, the process is at a (counterfactual) state $s$, and there are so far $l$ actions that have been different in comparison with the observed action sequence. Formally,

$$V_\tau(s, l, t) = \max_{a'_t, \ldots, a'_{T-1}} \sum_{t'=t}^{T-1} R(s'_{t'}, a'_{t'}) \qquad \text{subject to} \quad s'_t = s \text{ and } \sum_{t'=t}^{T-1} \mathbf{1}[a_{t'} \neq a'_{t'}] \leq k - l.$$

---

[5]The supporting proofs of all Theorems, Lemmas and Propositions can be found in Appendix D.

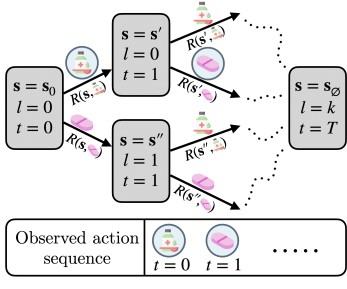

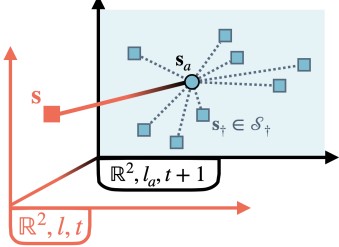

(a) Search graph  (b) Heuristic function computation

Figure 1: Main components of our search method based on the A* algorithm. Panel (a) shows the search graph for a problem instance with $|\mathcal{A}| = 2$. Here, each box represents a node $v = (\boldsymbol{s}, l, t)$ of the graph, and each edge represents a counterfactual transition. Next to each edge, we include the action $a \in \mathcal{A}$ causing the transition and the associated reward. Panel (b) shows the heuristic function computation, where the two axes represent a (continuous) state space $\mathcal{S} = \mathbb{R}^2$ and the two levels on the z-axis correspond to differences in the (integer) values $(l, t)$ and $(l_a, t + 1)$. Here, the blue squares correspond to the finite states in the anchor set $\mathcal{S}_\dagger$ and $(\boldsymbol{s}_a, l_a) = F_{\tau,t}^+ ((\boldsymbol{s}, l), a)$.

Then, it is easy to see that the quantity $V_\tau(\boldsymbol{s}, l, t)$ can be given by the recursive function

$$V_\tau(\boldsymbol{s}, l, t) = \max_{a \in \mathcal{A}} \left\{ R(\boldsymbol{s}, a) + V_\tau(\boldsymbol{s}_a, l_a, t + 1) \right\}, \quad \text{for all } \boldsymbol{s} \in \mathcal{S}, \ l < k \text{ and } t < T - 1, \quad (10)$$

where $(\boldsymbol{s}_a, l_a) = F_{\tau,t}^+ ((\boldsymbol{s}, l), a)$. In the base case of $l = k$ (*i.e.*, all allowed action changes are already performed), we have $V_\tau(\boldsymbol{s}, k, t) = R(\boldsymbol{s}, a_t) + V_\tau(\boldsymbol{s}_{a_t}, l_{a_t}, t + 1)$ for all $\boldsymbol{s} \in \mathcal{S}$ and $t < T - 1$, and $V_\tau(\boldsymbol{s}, k, T - 1) = R(\boldsymbol{s}, a_{T-1})$ for $t = T - 1$. Lastly, when $t = T - 1$ and $l < k$, we have $V_\tau(\boldsymbol{s}, l, T - 1) = \max_{a \in \mathcal{A}} R(\boldsymbol{s}, a)$ for all $\boldsymbol{s} \in \mathcal{S}$.

Given the optimal substructure of the problem, one may be tempted to employ a typical dynamic programming approach to compute the values $V_\tau(\boldsymbol{s}, l, t)$ in a bottom-up fashion. However, the complexity of the problem lies in the fact that, the states $\boldsymbol{s}$ are real-valued vectors whose exact values depend on the entire action sequence that led to them. Hence, to enumerate all the possible values that $\boldsymbol{s}$ might take, one has to enumerate all possible action sequences in the search space, which is equivalent to solving our problem with a brute force search. In what follows, we present our proposed method to find optimal solutions using the $A^*$ algorithm, with the caveat that its runtime varies depending on the problem instance, and it can be equal to that of a brute force search in the worst case.

**Casting the problem as graph search.** We represent the solution space of our problem as a graph, where each node $v$ corresponds to a tuple $(\boldsymbol{s}, l, t)$ with $\boldsymbol{s} \in \mathcal{S}$, $l \in [k]$ and $t \in [T]$. Every node $v = (\boldsymbol{s}, l, t)$ with $l < k$ and $t < T - 1$ has $|\mathcal{A}|$ outgoing edges, each one associated with an action $a \in \mathcal{A}$, carrying a reward $R(\boldsymbol{s}, a)$, and leading to a node $v_a = (\boldsymbol{s}_a, l_a, t + 1)$ such that $(\boldsymbol{s}_a, l_a) = F_{\tau,t}^+ ((\boldsymbol{s}, l), a)$. In the case of $l = k$, the node $v$ has exactly one edge corresponding to the observed action $a_t$ at time $t$. Lastly, when $t = T - 1$, the outgoing edge(s) lead(s) to a common node $v_T = (\boldsymbol{s}_\emptyset, k, T)$ which we call the *goal node*, and it has zero outgoing edges itself. Note that, the exact value of $\boldsymbol{s}_\emptyset$ is irrelevant, and we only include it for notational completeness.

Let $\boldsymbol{s}_0$ be the initial state of the observed episode. Then, it is easy to notice that, starting from the root node $v_0 = (\boldsymbol{s}_0, 0, 0)$, the first elements of each node $v_i$ on a path $v_0, \ldots, v_i, \ldots, v_T$ form a sequence of counterfactual states, and the edges that connect those nodes are such that the corresponding counterfactual action sequence differs from the observed one in at most $k$ actions. That said, the counterfactual outcome $o^+(\tau) = \sum_{t=0}^{T-1} R(\boldsymbol{s}_t', a_t')$ is expressed as the sum of the rewards associated with each edge in the path, and the problem defined by Eq. 9 is equivalent to finding the path of maximum total reward that starts from $v_0$ and ends in $v_T$. Figure 1a illustrates the search graph for a simple instance of our problem. Unfortunately, since the states $\boldsymbol{s}$ are vectors of real values, even enumerating all the graph's nodes requires time exponential in the number of actions $|\mathcal{A}|$, which makes classic algorithms that search over the entire graph non-practical.

To address this challenge, we resort to the $A^*$ algorithm, which performs a more efficient search over the graph by preferentially exploring only parts of it where we have prior information that they are

**Algorithm 1:** It computes upper bounds $\hat{V}_\tau(\boldsymbol{s}, l, t)$ for the values $V_\tau(\boldsymbol{s}, l, t)$

---

**Input**: States $\mathcal{S}$, actions $\mathcal{A}$, observed action sequence $\{a_t\}_{t=0}^{t=T-1}$, horizon $T$, transition function $F_{\tau,t}^+$,
 reward function $R$, constraint $k$, anchor set $\mathcal{S}_\dagger$.
**Initialize**: $\hat{V}_\tau(\boldsymbol{s}, l, T-1) \leftarrow \max_{a \in \mathcal{A}} R(\boldsymbol{s}, a), \quad s \in \mathcal{S}_\dagger, \; l = 0, \ldots, k-1$.
 $\qquad \hat{V}_\tau(\boldsymbol{s}, k, T-1) \leftarrow R(\boldsymbol{s}, a_t), \quad s \in \mathcal{S}_\dagger$.
**for** $t = T-2, \ldots, 0$ **do**
 **for** $l = k, \ldots, 0$ **do**
  $available\_actions \leftarrow a_t$ **if** $l = k$ **else** $\mathcal{A}$
  **for** $\boldsymbol{s} \in \mathcal{S}_\dagger$ **do**
   $bounds \leftarrow \emptyset$
   **for** $a \in available\_actions$ **do**
    $\boldsymbol{s}_a, l_a \leftarrow F_{\tau,t}^+((\boldsymbol{s}, l), a)$ ;  /* Get the min bound for $V_\tau(\boldsymbol{s}_a, l_a, t+1)$
    $V_a \leftarrow \min_{\boldsymbol{s}_\dagger \in \mathcal{S}_\dagger} \{\hat{V}_\tau(\boldsymbol{s}_\dagger, l_a, t+1) + L_{t+1} \|\boldsymbol{s}_\dagger - \boldsymbol{s}_a\|\}$ ;   based on $\mathcal{S}_\dagger$ */
    $bounds \leftarrow bounds \cup \{R(\boldsymbol{s}, a) + V_a\}$
   **end**
   $\hat{V}_\tau(\boldsymbol{s}, l, t) \leftarrow \max(bounds)$ ;   /* Get the max bound over the actions */
  **end**
 **end**
**end**
**return** $\hat{V}_\tau(\boldsymbol{s}, l, t)$  for all $\boldsymbol{s} \in \mathcal{S}_\dagger$, $l \in [k]$, $t \in [T-1]$

---

more likely to lead to paths of higher total reward. Concretely, the algorithm proceeds iteratively and maintains a queue of nodes to visit, initialized to contain only the root node $v_0$. Then, at each step, it selects one node from the queue, and it retrieves all its children nodes in the graph which are subsequently added to the queue. It terminates when the node being visited is the goal node $v_T$. Refer to Algorithm 2 in Appendix E for a pseudocode implementation of the $A^*$ algorithm.

The key element of the $A^*$ algorithm is the criterion based on which it selects which node from the queue to visit next. Let $v_i = (\boldsymbol{s}_i, l_i, t)$ be a candidate node in the queue and $r_{v_i}$ be the total reward of the path that the algorithm has followed so far to reach from $v_0$ to $v_i$. Then, the $A^*$ algorithm visits next the node $v_i$ that maximizes the sum $r_{v_i} + \hat{V}_\tau(\boldsymbol{s}_i, l_i, t)$, where $\hat{V}_\tau$ is a *heuristic function* that aims to estimate the maximum reward that can be achieved via any path starting from $v_i = (\boldsymbol{s}_i, l_i, t)$ and ending in the goal node $v_T$, *i.e.*, it gives an estimate for the quantity $V_\tau(\boldsymbol{s}_i, l_i, t)$. Intuitively, the heuristic function can be thought of as an "eye into the future" of the graph search, that guides the algorithm towards nodes that are more likely to lead to the optimal solution and the algorithm's performance depends on the quality of the approximation of $V_\tau(\boldsymbol{s}_i, l_i, t)$ by $\hat{V}_\tau(\boldsymbol{s}_i, l_i, t)$. Next, we will look for a heuristic function that satisfies *consistency*[6] to guarantee that the $A^*$ algorithm as described above returns the optimal solution upon termination [52].

**Computing a consistent heuristic function.** We first propose an algorithm that computes the function's values $\hat{V}_\tau(\boldsymbol{s}, l, t)$ for a finite set of points such that $l \in [k]$, $t \in [T-1]$, $\boldsymbol{s} \in \mathcal{S}_\dagger \subset \mathcal{S}$, where $\mathcal{S}_\dagger$ is a pre-defined *finite* set of states—an *anchor set*—whose construction we discuss later. Then, based on the Lipschitz-continuity of the SCM $\mathcal{C}$, we show that these computed values of $\hat{V}_\tau$ are valid upper bounds of the corresponding values $V_\tau(\boldsymbol{s}, l, t)$ and we expand the definition of the heuristic function $\hat{V}_\tau$ over all $\boldsymbol{s} \in \mathcal{S}$ by expressing it in terms of those upper bounds. Finally, we prove that the function resulting from the aforementioned procedure is consistent.

To compute the upper bounds $\hat{V}_\tau$, we exploit the observation that the values $V_\tau(\boldsymbol{s}, l, t)$ satisfy a form of Lipschitz-continuity, as stated in the following Lemma.

**Lemma 5.** *Let $\boldsymbol{u}_t = g_S^{-1}(\boldsymbol{s}_t, a_t, \boldsymbol{s}_{t+1})$, $K_{\boldsymbol{u}_t} = \max_{a \in \mathcal{A}} K_{a, \boldsymbol{u}_t}$, $C = \max_{a \in \mathcal{A}} C_a$ and the sequence $L_0, \ldots, L_{T-1} \in \mathbb{R}_+$ be such that $L_{T-1} = C$ and $L_t = C + L_{t+1} K_{\boldsymbol{u}_t}$ for $t \in [T-2]$. Then, it holds that $|V_\tau(\boldsymbol{s}, l, t) - V_\tau(\boldsymbol{s}', l, t)| \leq L_t \|\boldsymbol{s} - \boldsymbol{s}'\|$, for all $t \in [T-1]$, $l \in [k]$ and $\boldsymbol{s}, \boldsymbol{s}' \in \mathcal{S}$.*

Based on this observation, our algorithm proceeds in a bottom-up fashion and computes valid upper bounds of the values $V_\tau(\boldsymbol{s}, l, t)$ for all $l \in [k]$, $t \in [T-1]$ and $\boldsymbol{s}$ in the anchor set $\mathcal{S}_\dagger$. To get the

---

[6]A heuristic function $\hat{V}_\tau$ is consistent iff, for nodes $v = (\boldsymbol{s}, l, t)$, $v_a = (\boldsymbol{s}_a, l_a, t+1)$ connected with an edge associated with action $a$, it satisfies $\hat{V}_\tau(\boldsymbol{s}, l, t) \geq R(\boldsymbol{s}, a) + \hat{V}_\tau(\boldsymbol{s}_a, l_a, t+1)$ [53].

intuition, assume that, for a given $t$, the values $\hat{V}_\tau(s, l, t+1)$ are already computed for all $s \in \mathcal{S}_\dagger$, $l \in [k]$, and they are indeed valid upper bounds of the corresponding $V_\tau(s, l, t+1)$. Then, let $(s_a, l_a) = F^+_{\tau,t}((s, l), a)$ for some $s \in \mathcal{S}_\dagger$ and $l \in [k]$. Since $s_a$ itself may not belong to the finite anchor set $\mathcal{S}_\dagger$, the algorithm uses the values $\hat{V}_\tau(s_\dagger, l_a, t+1)$ of all anchors $s_\dagger \in \mathcal{S}_\dagger$ in combination with their distance to $s_a$, and it sets the value of $\hat{V}_\tau(s, l, t)$ in way that it is also guaranteed to be a (maximally tight) upper bound of $V_\tau(s, l, t)$. Figure 1b illustrates the above operation. Algorithm 1 summarizes the overall procedure, which is guaranteed to return upper bounds, as shown by the following proposition:

**Proposition 6.** *For all $s \in \mathcal{S}_\dagger$, $l \in [k], t \in [T-1]$, it holds that $\hat{V}_\tau(s, l, t) \geq V_\tau(s, l, t)$, where $\hat{V}_\tau(s, l, t)$ are the values of the heuristic function computed by Algorithm 1.*

Next, we use the values $\hat{V}_\tau(s, l, t)$ computed by Algorithm 1 to expand the definition of $\hat{V}_\tau$ over the entire domain as follows. For some $s \in \mathcal{S}$, $a \in \mathcal{A}$, let $(s_a, l_a) = F^+_{\tau,t}((s, l), a)$, then, we have that

$$\hat{V}_\tau(s, l, t) = \begin{cases} 0 & t = T \\ \max_{a \in \mathcal{A}'} R(s, a) & t = T-1 \\ \max_{a \in \mathcal{A}'} \left\{ R(s, a) + \min_{s_\dagger \in \mathcal{S}_\dagger} \left\{ \hat{V}_\tau(s_\dagger, l_a, t+1) + L_{t+1} \|s_\dagger - s_a\| \right\} \right\} & \text{otherwise,} \end{cases} \quad (11)$$

where $\mathcal{A}' = \{a_t\}$ for $l = k$ and $\mathcal{A}' = \mathcal{A}$ for $l < k$. Finally, the following theorem shows that the resulting heuristic function $\hat{V}_\tau$ is consistent:

**Theorem 7.** *For any nodes $v = (s, l, t), v_a = (s_a, l_a, t+1)$ with $t < T-1$ connected with an edge associated with action $a$, it holds that $\hat{V}_\tau(s, l, t) \geq R(s, a) + \hat{V}_\tau(s_a, l_a, t+1)$. Moreover, for any node $v = (s, l, T-1)$ and edge connecting it to the goal node $v_T = (s_\emptyset, k, T)$, it holds that $\hat{V}_\tau(s, l, T-1) \geq R(s, a) + \hat{V}_\tau(s_\emptyset, k, T)$.*

**Kick-starting the heuristic function computation with Monte Carlo anchor sets.** For any $s \notin \mathcal{S}_\dagger$, whenever we compute $\hat{V}_\tau(s, l, t)$ using Eq. 11, the resulting value is set based on the value $\hat{V}_\tau(s_\dagger, l_a, t+1)$ of some anchor $s_\dagger$, increased by a *penalty* term $L_{t+1} \|s_\dagger - s_a\|$. Intuitively, this allows us to think of the heuristic function $\hat{V}_\tau$ as an upper bound of the function $V_\tau$ whose looseness depends on the magnitude of the penalty terms encountered during the execution of Algorithm 1 and each subsequent evaluation of Eq. 11. To speed up the $A^*$ algorithm, note that, ideally, one would want all penalty terms to be zero, *i.e.*, an anchor set that includes all the states $s$ of the nodes $v = (s, l, t)$ that are going to appear in the search graph. However, as discussed in the beginning of Sec. 4, an enumeration of those states requires a runtime exponential in the number of actions.

To address this issue, we introduce a Monte Carlo simulation technique that adds to the anchor set the observed states $\{s_0, \ldots, s_{T-1}\}$ and all unique states $\{s'_0, \ldots, s'_{T-1}\}$ resulting by $M$ randomly sampled counterfactual action sequences $a'_0, \ldots, a'_{T-1}$. Specifically, for each action sequence, we first sample a number $k'$ of actions to be changed and what those actions are going to be, both uniformly at random from $\{1, \ldots, k\}$ and $\mathcal{A}^{k'}$, respectively. Then, we sample from $\{0, \ldots, T-1\}$ the $k'$ time steps where the changes take place, with each time step $t$ having a probability $L_t / \sum_{t'} L_{t'}$ to be selected. This biases the sampling towards earlier time steps, where the penalty terms are larger due to the higher Lipschitz constants. As we will see in the next section, this approach works well in practice, and it allows us to control the runtime of the $A^*$ algorithm by appropriately adjusting the number of samples $M$. We experiment with additional anchor set selection strategies in Appendix F.

## 5 Experiments using clinical sepsis management data

**Experimental setup.** To evaluate our method, we use real patient data from MIMIC-III [54], a freely accessible critical care dataset commonly used in reinforcement learning for healthcare [6, 55–57]. We follow the preprocessing steps of Komorowski et al. [6] to identify a cohort of 20,926 patients treated for sepsis [58]. Each patient record contains vital signs and administered treatment information in time steps of 4-hour intervals. As an additional preprocessing step, we discard patient records whose associated time horizon $T$ is shorter than 10, resulting in a final dataset of 15,992 patients with horizons between 10 and 20.

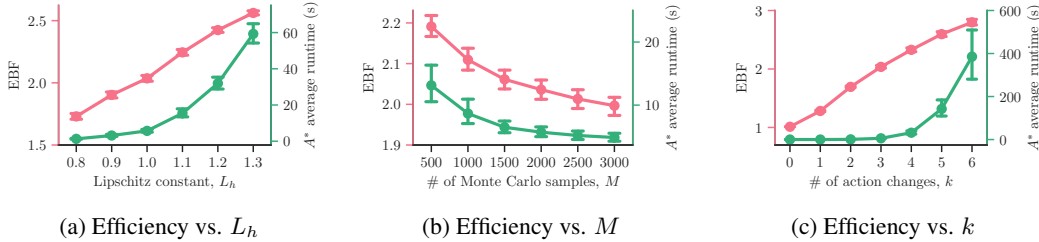

(a) Efficiency vs. $L_h$         (b) Efficiency vs. $M$         (c) Efficiency vs. $k$

Figure 2: Computational efficiency of our method under different configurations, as measured by the effective branching factor (pink-left axis) and the runtime of the $A^*$ algorithm (green-right axis). In Panel (a), we set $M = 2000$ and $k = 3$. In Panel (b), we set $L_h = 1.0$ and $k = 3$. In Panel (c), we set $L_h = 1.0$ and $M = 2000$. In all panels, we set $L_\phi = 0.1$ and error bars indicate $95\%$ confidence intervals over 200 executions of the $A^*$ algorithm for 200 patients with horizon $T = 12$.

To form our state space $\mathcal{S} = \mathbb{R}^D$, we use $D = 13$ features. Four of these features are demographic or contextual and thus we always set their counterfactual values to the observed ones. The remaining $\tilde{D} = 9$ features are time-varying and include the SOFA score [59]—a standardized score of organ failure rate—along with eight vital signs that are required for its calculation. Since SOFA scores positively correlate with patient mortality [60], we assume that each $\boldsymbol{s} \in \mathcal{S}$ gives a reward $R(\boldsymbol{s})$ equal to the negation of its SOFA value. Here, it is easy to see that this reward function is just a projection of $\boldsymbol{s}$, therefore, it is Lipschitz continuous with constant $C_a = 1$ for all $a \in \mathcal{A}$. Following related work [6, 55, 57], we consider an action space $\mathcal{A}$ that consists of 25 actions, which correspond to $5 \times 5$ levels of administered vasopressors and intravenous fluids. Refer to Appendix G for additional details on the features and actions.

To model the transition dynamics of the time-varying features, we consider an SCM $\mathcal{C}$ whose transition mechanism takes a location-scale form $g_S(\boldsymbol{S}_t, A_t, \boldsymbol{U}_t) = h(\boldsymbol{S}_t, A_t) + \phi(\boldsymbol{S}_t, A_t) \odot \boldsymbol{U}_t$, where $h, \phi : \mathcal{S} \times \mathcal{A} \to \mathbb{R}^{\tilde{D}}$, and $\odot$ denotes the element-wise multiplication [22, 24]. Notably, this model is element-wise bijective and hence it is counterfactually identifiable, as shown in Section 2. Moreover, we use neural networks to model the location and scale functions $h$ and $\phi$ and enforce their Lipschitz constants to be $L_h$ and $L_\phi$, respectively. This results in a Lipschitz continuous SCM $\mathcal{C}$ with $K_{a,\boldsymbol{u}} = L_h + L_\phi \max_i |u_i|$. Further, we assume that the noise variable $\boldsymbol{U}_t$ follows a multivariate Gaussian distribution with zero mean and allow its covariance matrix to be a (trainable) parameter.

We jointly train the weights of the networks $h$ and $\phi$ and the covariance matrix of the noise prior on the observed patient transitions using stochastic gradient descent with the negative log-likelihood of each transition as a loss. In our experiments, if not specified otherwise, we use an SCM with Lipschitz constants $L_h = 1.0$, $L_\phi = 0.1$ that achieves a log-likelihood only $6\%$ lower to that of the best model trained without any Lipschitz constraint. Refer to Appendix G for additional details on the network architectures, the training procedure and the way we enforce Lipschitz continuity.[7]

**Results.** We start by evaluating the computational efficiency of our method against (i) the Lipschitz constant of the location network $L_h$, (ii) the number of Monte Carlo samples $M$ used to generate the anchor set $\mathcal{S}_\dagger$, and (iii) the number of actions $k$ that can differ from the observed ones. We measure efficiency using running time and the effective branching factor (EBF) [52]. The EBF is defined as a real number $b \geq 1$ such that the number of nodes expanded by $A^*$ is equal to $1 + b + b^2 + \cdots + b^T$, where $T$ is the horizon, and values close to 1 indicate that the heuristic function is the most efficient in guiding the search. Figure 2 summarizes the results, which show that our method maintains overall a fairly low running time that decreases with the number of Monte Carlo samples $M$ used for the generation of the anchor set and increases with the Lipschitz constant $L_h$ and the number of action changes $k$. That may not come as a surprise since, as $L_h$ increases, the heuristic function becomes more loose, and as $k$ increases, the size of the search space increases exponentially. To put things in perspective, for a problem instance with $L_h = 1.0$, $k = 3$ and horizon $T = 12$, the $A^*$ search led by our heuristic function is effectively equivalent to an exhaustive search over a full tree with

---

[7]All experiments were performed using an internal cluster of machines equipped with 16 Intel(R) Xeon(R) 3.20GHz CPU cores, 512GBs of memory and 2 NVIDIA A40 48GB GPUs.

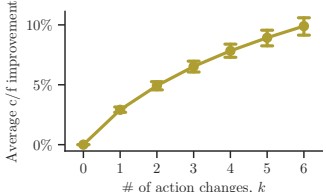

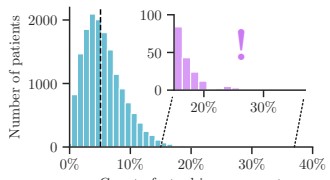

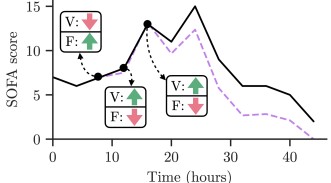

(a) Average c/f improvement vs. $k$     (b) Distribution of c/f improvement     (c) Observed vs. c/f episode

Figure 3: Retrospective analysis of patients' episodes. Panel (a) shows the average counterfactual improvement as a function of $k$ for a set of 200 patients with horizon $T = 12$, where error bars indicate 95% confidence intervals. Panel (b) shows the distribution of counterfactual improvement across all patients for $k = 3$, where the dashed vertical line indicates the median. Panel (c) shows the observed (solid) and counterfactual (dashed) SOFA score across time for a patient who presents a 19.9% counterfactual improvement when $k = 3$. Upward (downward) arrows indicate action changes that suggest a higher (lower) dosage of vasopressors (V) and fluids (F). In all panels, we set $M = 2000$.

$2.1^{12} \approx 7{,}355$ leaves while the corresponding search space of our problem consists of more than 3 million action sequences—more than 3 million paths to reach from the root node to the goal node.

Next, we investigate to what extent the counterfactual action sequences generated by our method would have led the patients in our dataset to better outcomes. For each patient, we measure their counterfactual improvement—the relative decrease in cumulative SOFA score between the counterfactual and the observed episode. Figures 3a and 3b summarize the results, which show that: (i) the average counterfactual improvement shows a diminishing increase as $k$ increases; (ii) the median counterfactual improvement is only 5%, indicating that, the treatment choices made by the clinicians for most of the patients were close to optimal, even with the benefit of hindsight; and (iii) there are 176 patients for whom our method suggests that a different sequence of actions would have led to an outcome that is at least 15% better. That said, we view patients at the tail of the distribution as "interesting cases" that should be deferred to domain experts for closer inspection, and we present one such example in Fig. 3c. In this example, our method suggests that, had the patient received an early higher dosage of intravenous fluids while some of the later administered fluids where replaced by vasopressors, their SOFA score would have been lower across time. Although we present this case as purely anecdotal, the counterfactual episode is plausible, since there are indications of decreased mortality when intravenous fluids are administered at the early stages of a septic shock [61].

## 6   Conclusions

In this paper, we have introduced the problem of finding counterfactually optimal action sequences in sequential decision making processes with continuous state dynamics. We showed that the problem is NP-hard and, to tackle it, we introduced a search method based on the $A^*$ algorithm that is guaranteed to find the optimal solution, with the caveat that its runtime can vary depending on the problem instance. Lastly, using real clinical data, we have found that our method is very efficient in practice, and it has the potential to offer interesting insights to domain experts by highlighting episodes and time-steps of interest for further inspection.

Our work opens up many interesting avenues for future work. For example, it would be interesting to develop algorithms with approximation guarantees that run in polynomial time, at the expense of not achieving strict counterfactual optimality. Moreover, since the practicality of methods like ours relies on the assumption that the SCM describing the environment is accurate, it would be interesting to develop methods to learn SCMs that align with human domain knowledge. Finally, it would be interesting to validate our method using real datasets from other applications and carry out user studies in which the counterfactual action sequences found by our method are systematically evaluated by the human experts (*e.g.*, clinicians) who took the observed actions.

**Acknowledgements.** Tsirtsis and Gomez-Rodriguez acknowledge support from the European Research Council (ERC) under the European Union's Horizon 2020 research and innovation programme (grant agreement No. 945719).

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

# A Further related work

**Counterfactual reasoning and reinforcement learning.** As mentioned in Section 1, there is a closely related line of work [10–13] that focuses on the development of machine learning methods that employ elements of counterfactual reasoning to improve or to retrospectively analyze decisions in sequential settings. Buesing et al. [10] use SCMs to express the transition dynamics in Partially Observable MDPs (POMDPs), and they propose a method to efficiently compute a policy based on counterfactual realizations of logged episodes. Lu et al. [12] adopt a similar modeling framework, and they propose a counterfactual data augmentation approach to speed up standard Q-learning. Oberst and Sontag [11] introduce the Gumbel-Max SCM to express the dynamics of an arbitrary discrete POMPD, and they develop a method for counterfactual off-policy evaluation to identify episodes where a given alternative policy would have achieved a higher reward. However, none of these works aims to find an action sequence, close to the observed sequence of a particular episode, that is counterfactually optimal.

**Planning in continuous-state MDPs.** Our work has additional connections to pieces of work that aim to approximate the optimal value function in an MDP with continuous states and a finite horizon [27, 62, 63]. Therein, the work most closely related to ours is the one by Bertsekas [27]. It shows that, under a Lipschitz continuity assumption on the transition dynamics, a value function computed via value iteration in an MDP with discretized states, converges to the optimal value function of the original (continous-state) MDP as the discretization becomes finer. Although some of the proof mechanics of this work are similar to ours, the contributions are orthogonal, as we do not employ any form of discretization, and we leverage the Lipschitz continuity assumption to compute optimal action sequences in continuous states using the $A^*$ algorithm.

**Counterfactual reasoning and explainability.** Our work has ties to pieces of work that use forms of counterfactual reasoning as a tool towards learning explainable machine learning models. For example, Madumal et al. [64] propose to express the action selection process of a reinforcement learning agent as a causal graph, and they use it to generate explanations for the agent's chosen actions. Bica et al. [65] introduce an inverse reinforcement learning approach to learn an interpretable reward function from expert demonstrated behavior that is expressed in terms of preferences over potential (counterfactual) outcomes. Moreover, our work is broadly related to the work on counterfactual explanations (in classification) that aims to find a minimally different set of features that would have led to a different outcome, in settings where single decisions are taken [66, 67].

# B    Supporting table and figure

Table 1: It summarizes the most important notation used in the main body of the paper.

| Symbol/Definition | Description |
|---|---|
| $T$ | Time horizon |
| $\boldsymbol{S}_t \in \mathcal{S}, A_t \in \mathcal{A}$ | State and action (random variables) at time $t$ |
| $\boldsymbol{s}_t, a_t$ | State and action (observed values) at time $t$ |
| $R : \mathcal{S} \times \mathcal{A} \to \mathbb{R}$ | Reward function |
| $\tau = \{(\boldsymbol{s}_t, a_t)\}_{t=0}^{T-1}$ | Observed episode |
| $o(\tau) = \sum_{t=0}^{T-1} R(\boldsymbol{s}_t, a_t)$ | Outcome of episode $\tau$ |
| $\boldsymbol{U}_t \in \mathcal{U}$ | Transition noise (random variable) at time $t$ |
| $\boldsymbol{u}_t$ | Transition noise (inferred value) at time $t$ |
| $g_S : \mathcal{S} \times \mathcal{A} \times \mathcal{U} \to \mathcal{S}$ | Transition mechanism (def. in Eq. 2) |
| $K_{a,\boldsymbol{u}}$ | Lipschitz constant of $g_S$ under $A_t = a$ and $\boldsymbol{U}_t = \boldsymbol{u}$ |
| $C_a$ | Lipschitz constant of $R$ under $A_t = a$ |
| $\mathcal{S}^+ = \mathcal{S} \times [T-1]$ | Enhanced state space |
| $(\boldsymbol{s}, l) \in \mathcal{S}^+$ | Enhanced state representing a c/f state $\boldsymbol{s}$ after $l$ action changes |
| $F_{\tau,t}^+ : \mathcal{S}^+ \times \mathcal{A} \to \mathcal{S}^+$ | Time-dependent transition function (def. in Eq. 7) |
| $\tau' = \{(\boldsymbol{s}_t', a_t')\}_{t=0}^{T-1}$ | Counterfactual episode |
| $o^+(\tau') = \sum_{t=0}^{T-1} R(\boldsymbol{s}_t', a_t')$ | Counterfactual outcome of episode $\tau'$ |
| $V_\tau(\boldsymbol{s}, l, t)$ | Max. c/f reward achievable for $\tau$ from enhanced state $(\boldsymbol{s}, l)$ at time $t$ |
| $(\boldsymbol{s}_a, l_a) = F_{\tau,t}^+((\boldsymbol{s}, l), a)$ | Enhanced state resulting from $(\boldsymbol{s}, l)$ with action $a$ |
| $v = (\boldsymbol{s}, l, t)$ | Node in the search graph of the $A^*$ algorithm |
| $v_0 = (\boldsymbol{s}, 0, 0)$ | Root node |
| $v_T = (\boldsymbol{s}_\emptyset, k, T)$ | Goal node |
| $r_v$ | Reward accumulated along the path that $A^*$ followed from $v_0$ to $v$ |
| $L_t$ | Lipschitz constant of $V_\tau(\boldsymbol{s}, l, t)$ |
| $\hat{V}_\tau(\boldsymbol{s}, l, t)$ | Heuristic function approximating $V_\tau(\boldsymbol{s}, l, t)$ |
| $\mathcal{S}_\dagger$ | Anchor set |

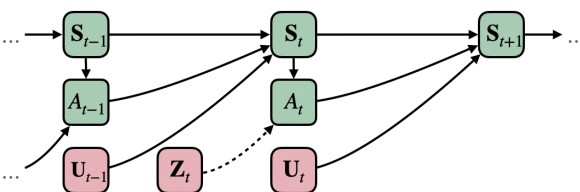

Figure 4: Directed acyclic graph $G$ of the SCM $\mathcal{C}$, modeling a sequential decision making process (figure adapted from [13]). Green boxes represent endogenous random variables and red boxes represent exogenous noise variables. All exogenous random variables are root nodes of the graph, and each one is independently sampled from its respective prior distribution. Each endogenous variable is an effect of its ancestors in the graph $G$, and takes its value based on Eqs. 1 and 2. An intervention $do(A_t = a)$ breaks the dependence of the variable $A_t$ from its ancestors (highlighted by dotted lines) and sets its value to $a$. After observing an event $\boldsymbol{S}_{t+1} = \boldsymbol{s}_{t+1}, \boldsymbol{S}_t = \boldsymbol{s}_t, A_t = a_t$, a counterfactual prediction corresponds to an intervention $do(A_t = a)$ in a modified SCM where $\boldsymbol{U}_t$ takes values $\boldsymbol{u}_t$ from a posterior distribution with support such that $\boldsymbol{s}_{t+1} = g_S(\boldsymbol{s}_t, a_t, \boldsymbol{u}_t)$.

# C   Counterfactual identifiability of element-wise bijective SCMs

In this section, we show that *element-wise bijective* SCMs, a subclass of bijective SCMs which we formally define next, are counterfactually identifiable.

**Definition 8.** *An SCM $\mathcal{C}$ is element-wise bijective iff it is bijective and there exist functions $g_{S,i}$ : $\mathbb{R} \times \mathcal{A} \times \mathbb{R} \to \mathbb{R}$ with $i \in \{1, \ldots, D\}$ such that, for every combination of $s_{t+1}, s_t, a_t, u_t$ with $s_{t+1} = g_S(s_t, a_t, u_t)$, it holds that $s_{t+1,i} = g_{S,i}(s_t, a_t, u_{t,i})$ for $i \in \{1, \ldots, D\}$.*

Under our assumption that the transition mechanism $g_S$ is continuous with respect to its third argument, it is easy to see that, for any element-wise bijective SCM, the functions $g_{S,i}$ are always strictly monotonic functions of the respective $u_{t,i}$. Based on this observation, we have the following theorem of counterfactual identifiability:

**Theorem 9.** *Let $\mathcal{C}$ and $\mathcal{M}$ be two element-wise bijective SCMs with transition mechanisms $g_S$ and $h_S$, respectively, and, for any observed transition $(s_t, a_t, s_{t+1})$, let $u_t = g_S^{-1}(s_t, a_t, s_{t+1})$ and $\tilde{u}_t = h_S^{-1}(s_t, a_t, s_{t+1})$. Moreover, given any $s \in \mathcal{S}, a \in \mathcal{A}$, let $s' = g_S(s, a, u_t)$ and $s'' = h_S(s, a, \tilde{u}_t)$. If $P^{\mathcal{C}}(S_{t+1} \mid S_t = s, A_t = a) = P^{\mathcal{M}}(S_{t+1} \mid S_t = s, A_t = a)$ for all $s \in \mathcal{S}, a \in \mathcal{A}$, it must hold that $s' = s''$.*

*Proof.* We prove the theorem by induction, starting by establishing the base case $s'_1 = s''_1$. Without loss of generality, assume that both $g_{S,1}$ and $h_{S,1}$ are strictly increasing with respect to their third argument. Since the two SCMs entail the same transition distributions, we have that

$$P^{\mathcal{C}}\left(S_{t+1,1} \leq s_{t+1,1} \mid S_t = s_t, A_t = a_t\right) = P^{\mathcal{M}}\left(S_{t+1,1} \leq s_{t+1,1} \mid S_t = s_t, A_t = a_t\right) \overset{(*)}{\Rightarrow}$$

$$P^{\mathcal{C}}\left(g_{S,1}\left(s_t, a_t, U_{t,1}\right) \leq g_{S,1}\left(s_t, a_t, u_{t,1}\right)\right) = P^{\mathcal{M}}\left(h_{S,1}\left(s_t, a_t, U_{t,1}\right) \leq h_{S,1}\left(s_t, a_t, \tilde{u}_{t,1}\right)\right) \overset{(**)}{\Rightarrow}$$

$$P^{\mathcal{C}}\left(U_{t,1} \leq u_{t,1}\right) = P^{\mathcal{M}}\left(U_{t,1} \leq \tilde{u}_{t,1}\right),$$

where $(*)$ holds because both SCMs are element-wise bijective, and $(**)$ holds because $g_{S,1}$ and $h_{S,1}$ are increasing with respect to their third argument. Similarly, we have that

$$
\begin{aligned}
P^{\mathcal{C}}\left(S_{t+1,1} \leq s'_1 \mid S_t = s, A_t = a\right) &= P^{\mathcal{C}}\left(g_{S,1}\left(s, a, U_{t,1}\right) \leq g_{S,1}\left(s, a, u_{t,1}\right)\right)\\
&\overset{(\star)}{=} P^{\mathcal{C}}\left(U_{t,1} \leq u_{t,1}\right)\\
&\overset{(\star\star)}{=} P^{\mathcal{M}}\left(U_{t,1} \leq \tilde{u}_{t,1}\right)\\
&\overset{(\dagger)}{=} P^{\mathcal{M}}\left(h_{S,1}\left(s, a, U_{t,1}\right) \leq h_{S,1}\left(s, a, \tilde{u}_{t,1}\right)\right)\\
&= P^{\mathcal{M}}\left(S_{t+1,1} \leq s''_1 \mid S_t = s, A_t = a\right)\\
&= P^{\mathcal{C}}\left(S_{t+1,1} \leq s''_1 \mid S_t = s, A_t = a\right),
\end{aligned}
$$

where in $(\star), (\dagger)$ we have used the monotonicity of $g_S$ and $h_S$, and $(\star\star)$ follows from the previous result. The last equality implies that $s'_1$ and $s''_1$ correspond to the same quantile of the distribution for $S_{t+1,1} \mid S_t = s, A_t = a$. Therefore, it is easy to see that $s'_1 = s''_1$ since the opposite would be in contradiction to $g_{S,1}$ being bijective. Note that, we can reach that conclusion irrespective of the direction of monotonicity of $g_{S,1}$ and $h_{S,1}$, since any change in the direction of the inequalities happening at step $(**)$ is reverted at steps $(\star)$ and $(\dagger)$.

Now, starting from the inductive hypothesis that $s'_i = s''_i$ for all $i \in \{1, \ldots, n\}$ with $n < D$, we show the inductive step, *i.e.*, $s'_{n+1} = s''_{n+1}$. Again, without loss of generality, assume that both $g_{S,n+1}$ and $h_{S,n+1}$ are strictly increasing with respect to their last argument. Note that, the two SCMs entail the same transition distributions, *i.e.*, the same joint distributions for $S_{t+1} \mid S_t, A_t$. Following from the law of total probability, they also entail the same conditional distributions for $S_{t+1,n+1} \mid S_{t+1,\leq n}, S_t, A_t$, where we use the notation $x_{\leq n}$ to refer to a vector that contains the first $n$ elements of a $D$-dimensional vector $x$. Therefore, we have that

$$P^{\mathcal{C}}\left(S_{t+1,n+1} \leq s_{t+1,n+1} \mid S_{t+1,\leq n} = s_{t+1,\leq n}, S_t = s_t, A_t = a_t\right) =$$

$$P^{\mathcal{M}}\left(S_{t+1,n+1} \leq s_{t+1,n+1} \mid S_{t+1,\leq n} = s_{t+1,\leq n}, S_t = s_t, A_t = a_t\right) \overset{(*)}{\Rightarrow}$$

$$P^{\mathcal{C}}\left(g_{S,n+1}\left(\boldsymbol{s}_t, a_t, U_{t,n+1}\right) \leq g_{S,n+1}\left(\boldsymbol{s}_t, a_t, u_{t,n+1}\right) \mid \boldsymbol{U}_{t,\leq n} = \boldsymbol{u}_{t,\leq n}\right)$$

$$= P^{\mathcal{M}}\left(h_{S,n+1}\left(\boldsymbol{s}_t, a_t, U_{t,n+1}\right) \leq h_{S,n+1}\left(\boldsymbol{s}_t, a_t, \tilde{u}_{t,n+1}\right) \mid \boldsymbol{U}_{t,\leq n} = \tilde{\boldsymbol{u}}_{t,\leq n}\right) \overset{(**)}{\Rightarrow}$$

$$P^{\mathcal{C}}\left(U_{t,n+1} \leq u_{t,n+1} \mid \boldsymbol{U}_{t,\leq n} = \boldsymbol{u}_{t,\leq n}\right) = P^{\mathcal{M}}\left(U_{t,n+1} \leq \tilde{u}_{t,n+1} \mid \boldsymbol{U}_{t,\leq n} = \tilde{\boldsymbol{u}}_{t,\leq n}\right),$$

where for the first equality we have used the inductive hypothesis, $(*)$ holds because both SCMs are element-wise bijective, and $(**)$ holds because $g_{S,n+1}$ and $h_{S,n+1}$ are increasing with respect to their third argument. Similarly, we get that

$$P^{\mathcal{C}}\left(S_{t+1,n+1} \leq s'_{n+1} \mid \boldsymbol{S}_{t+1,\leq n} = \boldsymbol{s}_{t+1,\leq n}, \boldsymbol{S}_t = \boldsymbol{s}, A_t = a\right)$$

$$= P^{\mathcal{C}}\left(g_{S,n+1}\left(\boldsymbol{s}, a, U_{t,n+1}\right) \leq g_{S,n+1}\left(\boldsymbol{s}, a, u_{t,n+1}\right) \mid g_{S,\leq n}\left(\boldsymbol{s}, a, \boldsymbol{U}_{t,\leq n}\right) = g_{s,\leq n}\left(\boldsymbol{s}, a, \boldsymbol{u}_{t,\leq n}\right)\right)$$

$$\overset{(\star)}{=} P^{\mathcal{C}}\left(U_{t,n+1} \leq u_{t,n+1} \mid \boldsymbol{U}_{t,\leq n} = \boldsymbol{u}_{t,\leq n}\right)$$

$$\overset{(\star\star)}{=} P^{\mathcal{M}}\left(U_{t,n+1} \leq \tilde{u}_{t,n+1} \mid \boldsymbol{U}_{t,\leq n} = \tilde{\boldsymbol{u}}_{t,\leq n}\right)$$

$$\overset{(\dagger)}{=} P^{\mathcal{M}}\left(h_{S,n+1}\left(\boldsymbol{s}, a, U_{t,1}\right) \leq h_{S,n+1}\left(\boldsymbol{s}, a, \tilde{u}_{t,1}\right) \mid h_{S,\leq n}\left(\boldsymbol{s}, a, \boldsymbol{U}_{t,\leq n}\right) = h_{s,\leq n}\left(\boldsymbol{s}, a, \tilde{\boldsymbol{u}}_{t,\leq n}\right)\right)$$

$$= P^{\mathcal{M}}\left(S_{t+1,n+1} \leq s''_{n+1} \mid \boldsymbol{S}_{t+1,\leq n} = \boldsymbol{S}_{t+1,\leq n}, \boldsymbol{S}_t = \boldsymbol{s}, A_t = a\right)$$

$$= P^{\mathcal{C}}\left(S_{t+1,n+1} \leq s''_{n+1} \mid \boldsymbol{S}_{t+1,\leq n} = \boldsymbol{s}_{t+1,\leq n}, \boldsymbol{S}_t = \boldsymbol{s}, A_t = a\right),$$

where in $(\star), (\dagger)$ we have used the invertibility and monotonicity of $g_S$ and $h_S$, and $(\star\star)$ follows from the previous result. With the same argument as in the base case, the last equality implies that $s'_{n+1} = s''_{n+1}$. That concludes the proof. $\qquad\square$

# D Proofs

## D.1 Proof of Theorem 4

We prove the hardness of our problem as defined in Eq. 9 by performing a reduction from the partition problem [26], which is known to be NP-Complete. In the partition problem, we are given a multiset of $B$ positive integers $\mathcal{V} = \{v_1, \ldots, v_B\}$ and the goal is to decide whether there is a partition of $\mathcal{V}$ into two subsets $\mathcal{V}_1, \mathcal{V}_2$ with $\mathcal{V}_1 \cap \mathcal{V}_2 = \emptyset$ and $\mathcal{V}_1 \cup \mathcal{V}_2 = \mathcal{V}$, such that their sums are equal, *i.e.*, $\sum_{v_i \in \mathcal{V}_1} v_i = \sum_{v_j \in \mathcal{V}_2} v_j$.

Consider an instance of our problem where $\mathcal{S} = \mathcal{U} = \mathbb{R}^2$, $\mathcal{A}$ contains 2 actions $a_{\mathrm{diff}}, a_{\mathrm{null}}$ and the horizon is $T = B + 1$. Let $\mathcal{C}$ be an element-wise bijective SCM with arbitrary prior distributions $P^{\mathcal{C}}(\boldsymbol{U}_t)$ such that their support is on $\mathbb{R}^2$ and a transition mechanism $g_S$ such that

$$g_S(\boldsymbol{S}_t, a_{\mathrm{diff}}, \boldsymbol{U}_t) = \begin{bmatrix} S_{t,1} - S_{t,2} \\ 0 \end{bmatrix} + \boldsymbol{U}_t \quad \text{and} \quad g_S(\boldsymbol{S}_t, a_{\mathrm{null}}, \boldsymbol{U}_t) = \begin{bmatrix} S_{t,1} \\ 0 \end{bmatrix} + \boldsymbol{U}_t. \tag{12}$$

Moreover, assume that the reward function is given by

$$
\begin{aligned}
R(\boldsymbol{S}_t, a_{\mathrm{diff}}) = R(\boldsymbol{S}_t, a_{\mathrm{null}}) = &- \max\left(0, S_{t,1} - \frac{sum(\mathcal{V})}{2} - S_{t,2}\frac{sum(\mathcal{V})}{2}\right) \\
&- \max\left(0, \frac{sum(\mathcal{V})}{2} - S_{t,1} - S_{t,2}\frac{sum(\mathcal{V})}{2}\right),
\end{aligned}
\tag{13}
$$

where $sum(\mathcal{V})$ is the sum of all elements $\sum_{i=1}^{B} v_i$. Note that, the SCM $\mathcal{C}$ defined above is Lipschitz-continuous as suggested by the following Lemma (refer to Appendix D.1.1 for a proof).

**Lemma 10.** *The SCM $\mathcal{C}$ defined by Equations 12, 13 is Lipschitz-continuous according to Definition 1.*

Now, assume that the counterfactual action sequence can differ in an arbitrary number of actions from the action sequence in the observed episode $\tau$, *i.e.*, $k = T$ and, let the observed action sequence be such that $a_t = a_{\mathrm{null}}$ for $t \in \{0, \ldots, T-1\}$. Lastly, let the initial observed state be $\boldsymbol{s}_0 = [0, v_1]$, the observed states $\{\boldsymbol{s}_t\}_{t=1}^{T-2}$ be such that $\boldsymbol{s}_t = \left[\sum_{i=1}^{t} v_i, v_{t+1}\right]$ for $t \in \{1, \ldots, T-2\}$ and the last observed state be $\boldsymbol{s}_{T-1} = [sum(\mathcal{V}), 0]$. Then, it is easy to see that the noise variables $\boldsymbol{U}_t$ have posterior distributions with a point mass on the respective values

$$\boldsymbol{u}_t = \begin{bmatrix} v_{t+1} \\ v_{t+2} \end{bmatrix} \text{ for } t \in \{0, \ldots, T-3\} \qquad \text{and} \qquad \boldsymbol{u}_{T-2} = \begin{bmatrix} v_{T-1} \\ 0 \end{bmatrix}.$$

Note that, for all $t \in \{1, \ldots, T-2\}$, we have $0 \le s_{t,1} < sum(\mathcal{V})$ and $s_{t,2} \ge 1$, hence the immediate reward according to Eq. 13 is equal to 0. Consequently, the outcome of the observed episode $\tau$ is $o^+(\tau) = R(\boldsymbol{s}_{T-1}, a_{\mathrm{null}}) = -\max(0, \frac{sum(\mathcal{V})}{2}) - \max(0, -\frac{sum(\mathcal{V})}{2}) = -\frac{sum(\mathcal{V})}{2}$.

Next, we will characterize the counterfactual outcome $o(\tau')$ of a counterfactual episode $\tau'$ with a sequence of states $\{\boldsymbol{s}_t'\}_{t=0}^{T-1}$ resulting from an alternative sequence of actions $\{a_t'\}_{t=0}^{T-1}$. Let $\mathcal{D}_t', \mathcal{N}_t'$ denote the set of time steps until time $t$, where the actions taken in a counterfactual episode $\tau'$ are $a_{\mathrm{diff}}$ and $a_{\mathrm{null}}$ respectively. Formally, $\mathcal{D}_t' = \{t' \in \{0, \ldots, t\} : a_{t'}' = a_{\mathrm{diff}}\}$, $\mathcal{N}_t' = \{t' \in \{0, \ldots, t\} : a_{t'}' = a_{\mathrm{null}}\}$. Then, as an intermediate result, we get the following Lemma (refer to Appendix D.1.2 for a proof).

**Lemma 11.** *It holds that $s_{t,1}' = \sum_{t' \in \mathcal{N}_{t-1}'} v_{t'+1}$ for all $t \in \{1, \ldots T-1\}$.*

Following from that, we get that $0 \le s_{t,1}' \le sum(\mathcal{V})$ for all $t \in \{1, \ldots, T-1\}$. Moreover, we can observe that the transition mechanism given in Eq. 12 is such that $g_{S,2}(\boldsymbol{S}_t, A_t, U_{t,2}) = U_{t,2}$ for all $t \in \{0, \ldots, T-2\}$, independently of $\boldsymbol{S}_T$ and $A_t$. Therefore, it holds that $s_{t,2}' = u_{t-1,2} \ge 1$ for $t \in \{1, \ldots, T-2\}$, and $s_{0,2}' = s_{0,2} = v_1 \ge 1$. As a direct consequence, it is easy to see that $R(\boldsymbol{s}_t', a_t') = 0$ for all $t \in \{0, \ldots, T-2\}$, and the counterfactual outcome is given by

$$o^+(\tau') = R(\boldsymbol{s}_{T-1}', a_{T-1}'), \tag{14}$$

In addition to that, we have that $u_{T-2,2} = 0$, hence

$$\boldsymbol{s}_{T-1}' = \begin{bmatrix} \sum_{t \in \mathcal{N}_{T-2}'} v_{t+1} \\ 0 \end{bmatrix} \tag{15}$$

Now, we will show that, if we can find the action sequence $\{a_t^*\}_{t=0}^{T-1}$ that gives the optimal coun­terfactual outcome $o^+(\tau^*)$ for the aforementioned instance in polynomial time, then we can make a decision about the corresponding instance of the partition problem, also in polynomial time. To this end, let $\{s_t^*\}_{t=0}^{T-1}$ be the sequence of states in the optimal counterfactual realization and, let $\mathcal{D}_{T-2}^* = \{t \in \{0, \ldots, T-2\} : a_t^* = a_{\mathrm{diff}}\}, \mathcal{N}_{T-2}^* = \{t' \in \{0, \ldots, T-2\} : a_{t'}^* = a_{\mathrm{null}}\}$.

From Eq. 14, we get that the optimal counterfactual outcome is $o^+(\tau^*) = R(s_{T-1}^*, a_{T-1}^*)$, and it is easy to see that the reward function given in Eq. 13 is always less or equal than zero. If $o(\tau^*) = 0$, it has to hold that

$$\max\left(0, s_{T-1,1}^* - \frac{sum(\mathcal{V})}{2} - s_{T-1,2}^* \frac{sum(\mathcal{V})}{2}\right) =$$

$$\max\left(0, \frac{sum(\mathcal{V})}{2} - s_{T-1,1}^* - s_{T-1,2}^* \frac{sum(\mathcal{V})}{2}\right) = 0 \overset{(*)}{\Rightarrow}$$

$$\left(\sum_{t \in \mathcal{N}_{T-2}^*} v_{t+1}\right) - \frac{sum(\mathcal{V})}{2} \leq 0 \quad \text{and} \quad \frac{sum(\mathcal{V})}{2} - \left(\sum_{t \in \mathcal{N}_{T-2}^*} v_{t+1}\right) \leq 0 \Rightarrow$$

$$\sum_{t \in \mathcal{N}_{T-2}^*} v_{t+1} = \frac{sum(\mathcal{V})}{2},$$

where $(*)$ follows from Eq. 15. As a consequence, the subsets $\mathcal{V}_1 = \{v_i : i - 1 \in \mathcal{N}_{T-2}^*\}$ and $\mathcal{V}_2 = \{v_i : i - 1 \in \mathcal{D}_{T-2}^*\}$ partition $\mathcal{V}$ and their sums are equal.

On the other hand, if $o^+(\tau^*) < 0$, as we will show, there is no partition of $\mathcal{V}$ into two sets with equal sums. For the sake of contradiction, assume there are two sets $\mathcal{V}_1, \mathcal{V}_2$ that partition $\mathcal{V}$, with $sum(\mathcal{V}_1) = sum(\mathcal{V}_2) = sum(\mathcal{V})/2$, and let $\mathcal{N}_{T-2}' = \{t \in \{0, \ldots, T-2\} : v_{t+1} \in \mathcal{V}_1\}$ and $\mathcal{D}_{T-2}' = \{t \in \{0, \ldots, T-2\} : v_{t+1} \in \mathcal{V}_2\}$. Then, consider the counterfactual episode $\tau'$ with an action sequence $\{a_t'\}_{t=0}^{T-1}$ such that its elements take values $a_{\mathrm{null}}$ and $a_{\mathrm{diff}}$ based on the sets $\mathcal{N}_{T-2}', \mathcal{D}_{T-2}'$ respectively, with $a_{T-1}'$ taking an arbitrary value. It is easy to see that

$$o^+(\tau') = R(s_{T-1}', a_{T-1}')$$

$$= R\left(\begin{bmatrix} \sum_{t \in \mathcal{N}_{T-2}'} v_{t+1} \\ 0 \end{bmatrix}, a_{T-1}'\right)$$

$$= -\max\left(0, \sum_{t \in \mathcal{N}_{T-2}'} v_{t+1} - \frac{sum(\mathcal{V})}{2}\right) - \max\left(0, \frac{sum(\mathcal{V})}{2} - \sum_{t \in \mathcal{N}_{T-2}'} v_{t+1}\right)$$

$$= -\max\left(0, \frac{sum(\mathcal{V})}{2} - \frac{sum(\mathcal{V})}{2}\right) - \max\left(0, \frac{sum(\mathcal{V})}{2} - \frac{sum(\mathcal{V})}{2}\right)$$

$$= 0 > o^+(\tau^*),$$

which is a contradiction. This step concludes the reduction and, therefore, the problem given in Eq. 9 cannot be solved in polynomial time, unless $P = NP$.

### D.1.1 Proof of Lemma 10

It is easy to see that, for all $u \in \mathcal{U}$ and for all $s, s' \in \mathcal{S}$, the function $g_S(S_t, a_{\mathrm{null}}, u)$ satisfies $\|g_S(s, a_{\mathrm{null}}, u) - g_S(s', a_{\mathrm{null}}, u)\| \leq \|s - s'\|$, and therefore $K_{a_{\mathrm{null}}, u} = 1$ satisfies Definition 1. For the case of $A_t = a_{\mathrm{diff}}$, we have that

$$\|g_S(s, a_{\mathrm{diff}}, u) - g_S(s', a_{\mathrm{diff}}, u)\| = \left\|\begin{bmatrix} s_1 - s_2 \\ 0 \end{bmatrix} - \begin{bmatrix} s_1' - s_2' \\ 0 \end{bmatrix}\right\| = \left\|\begin{bmatrix} (s_1 - s_1') + (s_2' - s_2) \\ 0 \end{bmatrix}\right\|$$

$$= |(s_1 - s_1') + (s_2' - s_2)| \leq |s_1 - s_1'| + |s_2 - s_2'|,$$

and therefore $\|g_S(s, a_{\mathrm{diff}}, u) - g_S(s', a_{\mathrm{diff}}, u)\|^2 \leq (s_1 - s_1')^2 + (s_2 - s_2')^2 + 2|s_1 - s_1'||s_2 - s_2'|$. We also have that

$$\sqrt{2}\|s - s'\| = \sqrt{2}\sqrt{(s_1 - s_1')^2 + (s_2 - s_2')^2} \Rightarrow 2\|s - s'\|^2 = 2(s_1 - s_1')^2 + 2(s_2 - s_2')^2.$$

By combining these, we get

$$2\left\Vert\boldsymbol{s}-\boldsymbol{s}'\right\Vert^2-\left\Vert g_S(\boldsymbol{s},a_{\text{diff}},\boldsymbol{u})-g_S(\boldsymbol{s}',a_{\text{diff}},\boldsymbol{u})\right\Vert^2\geq(s_1-s_1')^2+(s_2-s_2')^2-2|s_1-s_1'||s_2-s_2'|\Rightarrow$$
$$2\left\Vert\boldsymbol{s}-\boldsymbol{s}'\right\Vert^2-\left\Vert g_S(\boldsymbol{s},a_{\text{diff}},\boldsymbol{u})-g_S(\boldsymbol{s}',a_{\text{diff}},\boldsymbol{u})\right\Vert^2\geq(|s_1-s_1'|-|s_2-s_2'|)^2\geq0.$$

Hence, we can easily see that $K_{a_{\text{diff}},\boldsymbol{u}}=\sqrt{2}$ satisfies Definition 1.

Next, we need to show that, for all $a\in\mathcal{A}$ there exists a $C_a\in\mathbb{R}_+$ such that, for all $\boldsymbol{s},\boldsymbol{s}'\in\mathcal{S}$, it holds $|R(\boldsymbol{s},a)-R(\boldsymbol{s}',a)|\leq C_a\left\Vert\boldsymbol{s}-\boldsymbol{s}'\right\Vert$. Note that, to show that a function of the form $\max(0,f(\boldsymbol{s}))$ with $f:\mathbb{R}^2\to\mathbb{R}$ is Lipschitz continuous, it suffices to show that $f(\boldsymbol{s})$ is Lipschitz continuous, since the function $\max(0,x)$ with $x\in\mathbb{R}$ has a Lipschitz constant equal to 1.

We start by showing that the function $f(\boldsymbol{s})=s_1-\alpha-s_2\cdot\alpha$ is Lipschitz continuous, where $\alpha=sum(\mathcal{V})/2$ is a positive constant. For an arbitrary pair $\boldsymbol{s},\boldsymbol{s}'\in\mathcal{S}$, we have that

$$|f(\boldsymbol{s})-f(\boldsymbol{s}')|=|s_1-s_1'-\alpha(s_2-s_2')|\leq|s_1-s_1'|+\alpha|s_2-s_2'|\Rightarrow$$
$$|f(\boldsymbol{s})-f(\boldsymbol{s}')|^2\leq(s_1-s_1')^2+(s_2-s_2')^2+2\alpha|s_1-s_1'||s_2-s_2'|.$$

We also have that

$$\sqrt{1+\alpha}\left\Vert\boldsymbol{s}-\boldsymbol{s}'\right\Vert=\sqrt{1+\alpha}\sqrt{(s_1-s_1')^2+(s_2-s_2')^2}\Rightarrow$$
$$(1+\alpha)\left\Vert\boldsymbol{s}-\boldsymbol{s}'\right\Vert^2=(1+\alpha)(s_1-s_1')^2+(1+\alpha)(s_2-s_2')^2$$

By combining these, we get

$$(1+\alpha)\left\Vert\boldsymbol{s}-\boldsymbol{s}'\right\Vert^2-|f(\boldsymbol{s})-f(\boldsymbol{s}')|^2\geq\alpha(s_1-s_1')^2+\alpha(s_2-s_2')^2-2\alpha|s_1-s_1'||s_2-s_2'|\Rightarrow$$
$$(1+\alpha)\left\Vert\boldsymbol{s}-\boldsymbol{s}'\right\Vert^2-|f(\boldsymbol{s})-f(\boldsymbol{s}')|^2\geq\alpha(|s_1-s_1'|-|s_2-s_2'|)^2\geq0.$$

Hence, we arrive to $|f(\boldsymbol{s})-f(\boldsymbol{s}')|\leq\sqrt{1+\alpha}\left\Vert\boldsymbol{s}-\boldsymbol{s}'\right\Vert$, and the function $f$ is Lipschitz continuous. It is easy to see that the function $\phi(\boldsymbol{s})=\alpha-s_1-s_2\cdot\alpha$ is also Lipschitz continuous with the proof being almost identical. As a direct consequence, the reward function given in Equation 13 satisfies Definition 1 with $C_{a_{\text{null}}}=C_{a_{\text{diff}}}=2\sqrt{1+\frac{sum(\mathcal{V})}{2}}$. This concludes the proof of the lemma.

### D.1.2    Proof of Lemma 11

We will prove the lemma by induction. For the base case of $t=1$, we distinguish between the cases $a_0'=a_{\text{diff}}$ and $a_0'=a_{\text{null}}$. In the first case, we have $s_{1,1}'=u_{0,1}+s_{0,1}-s_{0,2}=v_1+0-v_1=0$ and $\mathcal{N}_0'=\emptyset$ and, therefore, the statement holds. In the second case, we have $s_{1,1}'=u_{0,1}+s_{0,1}=v_1+0=v_1$, $\mathcal{N}_0'=\{0\}$ and $\sum_{t'\in\mathcal{N}_0'}v_{t'+1}=v_1$. Therefore, the statement also holds.

For the inductive step ($t>1$), we assume that $s_{t-1,1}'=\sum_{t'\in\mathcal{N}_{t-2}'}v_{t'+1}$ and we will show that $s_{t,1}'=\sum_{t'\in\mathcal{N}_{t-1}'}v_{t'+1}$. Again, we distinguish between the cases $a_{t-1}'=a_{\text{diff}}$ and $a_{t-1}'=a_{\text{null}}$. However, note that, in both cases, $s_{t-1,2}'=u_{t-2,2}+0=v_t$. Therefore, in the case of $a_{t-1}'=a_{\text{diff}}$, we get

$$s_{t,1}'=u_{t-1,1}+s_{t-1,1}'-s_{t-1,2}'=v_t+\sum_{t'\in\mathcal{N}_{t-2}'}v_{t'+1}-v_t=\sum_{t'\in\mathcal{N}_{t-2}'}v_{t'+1}=\sum_{t'\in\mathcal{N}_{t-1}'}v_{t'+1},$$

where the last equation holds because $a_{t-1}'=a_{\text{diff}}$ and, therefore, $\mathcal{N}_{t-1}'=\mathcal{N}_{t-2}'$. In the case of $a_{t-1}'=a_{\text{null}}$, we get

$$s_{t,1}'=u_{t-1,1}+s_{t-1,1}'=v_t+\sum_{t'\in\mathcal{N}_{t-2}'}v_{t'+1}=\sum_{t'\in\mathcal{N}_{t-1}'}v_{t'+1},$$

where the last equation holds because $a_{t-1}'=a_{\text{null}}$ and, therefore, $\mathcal{N}_{t-1}'=\mathcal{N}_{t-2}'\cup\{t-1\}$.

### D.2    Proof of Lemma 5

We will prove the proposition by induction, starting from the base case, where $t=T-1$. First, Let $t=T-1$ and $l=k$. It is easy to see that, if the process is at a state $\boldsymbol{s}\in\mathcal{S}$ in the last

time step with no action changes left, the best reward that can be achieved is $R(s, a_{T-1})$, as already discussed after Eq. 10. Therefore, it holds that $|V_\tau(s, k, T-1) - V_\tau(s', k, T-1)| = |R(s, a_{T-1}) - R(s', a_{T-1})| \le C_{a_{T-1}} \|s - s'\| \le C \|s - s'\|$, where the last step holds because $C = \max_{a \in \mathcal{A}} C_a$. Now, consider the case of $t = T - 1$ with $l$ taking an arbitrary value in $\{0, \ldots, k-1\}$. Let $s, s'$ be two states in $\mathcal{S}$ and $a^*$ be the action that gives the maximum immediate reward at state $s$, that is, $a^* = \operatorname{argmax}_{a \in \mathcal{A}}\{R(s, a)\}$. Then, we get

$$
|V_\tau(s, l, T-1) - V_\tau(s', l, T-1)| = |\max_{a \in \mathcal{A}}\{R(s, a)\} - \max_{a \in \mathcal{A}}\{R(s', a)\}|
$$

$$
\overset{(*)}{\le} |R(s, a^*) - R(s', a^*)| \le C_{a^*} \|s - s'\| \le C \|s - s'\|,
$$

where $(*)$ follows from the fact that $R(s', a^*) \le \max_{a \in \mathcal{A}}\{R(s', a)\}$. Therefore, for any $l \in \{0, \ldots, k\}$ and $s, s' \in \mathcal{S}$, it holds that $|V_\tau(s, l, T-1) - V_\tau(s', l, T-1)| \le L_{T-1} \|s - s'\|$, where $L_{T-1} = C$.

Now, we will proceed to the induction step. Let $t < T - 1$, $l < k$ and, as an inductive hypothesis, assume that $L_{t+1} \in \mathbb{R}_+$ as defined in Lemma 5 is such that, for all $l \in \{0, \ldots, k\}$ and $s, s' \in \mathcal{S}$, it holds that $|V_\tau(s, l, t+1) - V_\tau(s', l, t+1)| \le L_{t+1} \|s - s'\|$. Additionally, let $(s_a, l_a), (s'_a, l_a)$ denote the enhanced states that follow from $(s, l), (s', l)$ after taking an action $a$, i.e., $(s_a, l_a) = F^+_{\tau, t}((s, l), a)$ and $(s'_a, l_a) = F^+_{\tau, t}((s', l), a)$. Lastly, let $a^*$ be the action that maximizes the future total reward starting from state $s$, i.e., $a^* = \operatorname{argmax}_{a \in \mathcal{A}}\{R(s, a) + V_\tau(s_a, l_a, t+1)\}$. Then, we have that

$$
|V_\tau(s, l, t) - V_\tau(s', l, t)|
$$

$$
= |\max_{a \in \mathcal{A}}\{R(s, a) + V_\tau(s_a, l_a, t+1)\} - \max_{a \in \mathcal{A}}\{R(s', a) + V_\tau(s'_a, l_a, t+1)\}|
$$

$$
\overset{(*)}{\le} |R(s, a^*) + V_\tau(s_{a^*}, l_{a^*}, t+1) - R(s', a^*) - V_\tau(s'_{a^*}, l_{a^*}, t+1)|
$$

$$
\le |R(s, a^*) - R(s', a^*)| + |V_\tau(s_{a^*}, l_{a^*}, t+1) - V_\tau(s'_{a^*}, l_{a^*}, t+1)|
$$

$$
\overset{(**)}{\le} C_{a^*} \|s - s'\| + L_{t+1} \|s_{a^*} - s'_{a^*}\|
$$

$$
\le C_{a^*} \|s - s'\| + L_{t+1} K_{a^*, \boldsymbol{u}_t} \|s - s'\|
$$

$$
\overset{(***)}{\le} C \|s - s'\| + L_{t+1} K_{\boldsymbol{u}_t} \|s - s'\|
$$

$$
= (C + L_{t+1} K_{\boldsymbol{u}_t}) \|s - s'\| = L_t \|s - s'\|.
$$

In the above, $(*)$ holds due to $R(s', a^*) + V_\tau(s'_{a^*}, l_{a^*}, t+1) \le \max_{a \in \mathcal{A}}\{R(s', a) + V_\tau(s'_a, l_a, t+1)\}$, $(**)$ follows from the inductive hypothesis, and $(***)$ holds because $C = \max_{a \in \mathcal{A}} C_a$ and $K_{\boldsymbol{u}_t} = \max_{a \in \mathcal{A}} K_{a, \boldsymbol{u}_t}$. It is easy to see that, similar arguments hold for the simple case of $l = k$, therefore, we omit the details. This concludes the inductive step and the proof of Lemma 5.

### D.3 Proof of Proposition 6

We will prove the proposition by induction, starting from the base case, where $t = T - 1$. If $t = T - 1$, the algorithm initializes $\hat{V}_\tau(s, l, T-1)$ to $\max_{a \in \mathcal{A}} R(s, a)$ for all $s \in \mathcal{S}_\dagger$, $l \in \{0, \ldots, k-1\}$ and $\hat{V}_\tau(s, k, T-1)$ to $R(s, a_{T-1})$. It is easy to see that those values are optimal, as already discussed after Eq. 10. Therefore, the base case $\hat{V}_\tau(s, l, T-1) \ge V_\tau(s, l, T-1)$ follows trivially.

Now, we will proceed to the induction step. Let $t < T - 1$ and, as an inductive hypothesis, assume that $\hat{V}_\tau(s, l, t+1) \ge V_\tau(s, l, t+1)$ for all $s \in \mathcal{S}_\dagger$, $l \in \{0, \ldots, k\}$. Our goal is to show that $\hat{V}_\tau(s, l, t) \ge V_\tau(s, l, t)$ for all $s \in \mathcal{S}_\dagger$, $l \in \{0, \ldots, k\}$. First, let $l < k$. For a given point $s \in \mathcal{S}_\dagger$, Algorithm 1 finds the next state $s_a$ that would have occurred by taking each action $a$, i.e., $(s_a, l_a) = F^+_{\tau, t}((s, l), a)$, and it computes the associated value $V_a = \min_{s_\dagger \in \mathcal{S}_\dagger}\{\hat{V}_\tau(s_\dagger, l_a, t+1) + L_{t+1} \|s_\dagger - s_a\|\}$. Then, it

simply sets $\hat{V}_\tau(s, l, t)$ equal to $\max_{a\in\mathcal{A}}\{R(s, a) + V_a\}$. We have that

$$
\begin{aligned}
V_a &= \min_{s_\dagger\in\mathcal{S}_\dagger}\{\hat{V}_\tau(s_\dagger, l_a, t+1) + L_{t+1}\|s_\dagger - s_a\|\} \\
&\overset{(*)}{\geq} \min_{s_\dagger\in\mathcal{S}_\dagger}\{V_\tau(s_\dagger, l_a, t+1) + L_{t+1}\|s_\dagger - s_a\|\} \\
&\overset{(**)}{\geq} \min_{s_\dagger\in\mathcal{S}_\dagger}\{V_\tau(s_a, l_a, t+1)\} \\
&= V_\tau(s_a, l_a, t+1),
\end{aligned}
$$

where $(*)$ follows from the inductive hypothesis, and $(**)$ is a consequence of Lemma 5. Then, we get

$$
\hat{V}_\tau(s, l, t) = \max_{a\in\mathcal{A}}\{R(s, a) + V_a\} \geq \max_{a\in\mathcal{A}}\{R(s, a) + V_\tau(s_a, l_a, t+1)\} = V_\tau(s, l, t).
$$

Additionally, when $l = k$, we have $\hat{V}_\tau(s, k, t) = R(s, a_t) + \min_{s_\dagger\in\mathcal{S}_\dagger}\{\hat{V}_\tau(s_\dagger, k, t+1) + L_{t+1}\|s_\dagger - s_{a_t}\|\}$ and $V_\tau(s, k, t) = R(s, a_t) + V_\tau(s_{a_t}, k, t+1)$. Therefore, the proof for $\hat{V}_\tau(s, k, t) \geq V_\tau(s, k, t)$ is almost identical.

## D.4 Proof of Theorem 7

We start from the case where $t = T - 1$. Let $v = (s, l, T-1)$ and, consider an edge associated with action $a^*$ connecting $v$ to the goal node $v_T = (s_\emptyset, k, T)$ that carries a reward $R(s, a^*)$. Then, we have

$$
\hat{V}_\tau(s, l, T-1) = \max_{a\in\mathcal{A}'}R(s, a) \geq R(s, a^*) + 0 = R(s, a^*) + \hat{V}_\tau(s_\emptyset, k, T),
$$

and the base case holds.

For the more general case, where $t < T - 1$, we first establish the following intermediate result, whose proof is given in Appendix D.4.1.

**Lemma 12.** *For every $s, s' \in \mathcal{S}$, $l \in \{0, \ldots, k\}$, $t \in \{0, \ldots, T-1\}$, it holds that $|\hat{V}_\tau(s, l, t) - \hat{V}_\tau(s', l, t)| \leq L_t\|s - s'\|$, where $L_t$ is as defined in Lemma 5.*

That said, consider an edge associated with an action $a^*$ connecting node $v = (s, l, t)$ to node $v_{a^*} = (s_{a^*}, l_{a^*}, t+1)$. Then, we have

$$
\begin{aligned}
\hat{V}_\tau(s, l, t) &= \max_{a\in\mathcal{A}'}\left\{R(s, a) + \min_{s_\dagger\in\mathcal{S}_\dagger}\left\{\hat{V}_\tau(s_\dagger, l_a, t+1) + L_{t+1}\|s_\dagger - s_a\|\right\}\right\} \\
&\geq R(s, a^*) + \min_{s_\dagger\in\mathcal{S}_\dagger}\left\{\hat{V}_\tau(s_\dagger, l_{a^*}, t+1) + L_{t+1}\|s_\dagger - s_{a^*}\|\right\} \\
&\geq R(s, a^*) + \min_{s_\dagger\in\mathcal{S}_\dagger}\left\{\hat{V}_\tau(s_{a^*}, l_{a^*}, t+1)\right\} \\
&= R(s, a^*) + \hat{V}_\tau(s_{a^*}, l_{a^*}, t+1).
\end{aligned}
$$

That concludes the proof and, therefore, the heuristic function $\hat{V}_\tau$ is consistent.

## D.4.1 Proof of Lemma 12

Without loss of generality, we will assume that $l < k$, since the proof for the case of $l = k$ is similar and more straightforward. We start from the case where $t = T - 1$ and, for two states $s, s' \in \mathcal{S}$ we have

$$
\begin{aligned}
|\hat{V}_\tau(s, l, T-1) - \hat{V}_\tau(s', l, T-1)| &= \left|\max_{a\in\mathcal{A}}R(s, a) - \max_{a\in\mathcal{A}}R(s', a)\right| \\
&= |V_\tau(s, l, T-1) - V_\tau(s', l, T-1)| \leq C\|s - s'\| = L_{T-1}\|s - s'\|,
\end{aligned}
$$

where the last inequality follows from Lemma 5.

Now, consider the case $t < T - 1$, and let $(s_a, l_a)$ denote the enhanced state that follows from $(s, l)$ after taking an action $a$ at time $t$, *i.e.*, $(s_a, l_a) = F_{\tau,t}^+((s, l), a)$. Then, we have

$$
|\hat{V}_\tau(s, l, t) - \hat{V}_\tau(s', l, t)|
$$

$$
= \left| \max_{a \in \mathcal{A}} \left\{ R(s, a) + \min_{s_\dagger \in \mathcal{S}_\dagger} \left\{ \hat{V}_\tau(s_\dagger, l_a, t+1) + L_{t+1} \|s_\dagger - s_a\| \right\} \right\} \right.
$$

$$
\left. - \max_{a \in \mathcal{A}} \left\{ R(s', a) + \min_{s_\dagger \in \mathcal{S}_\dagger} \left\{ \hat{V}_\tau(s_\dagger, l_a, t+1) + L_{t+1} \|s_\dagger - s'_a\| \right\} \right\} \right|. \quad (16)
$$

Let $a^*$ be the action $a \in \mathcal{A}$ that maximizes the first part of the above subtraction, *i.e.*,

$$
a^* = \operatorname*{argmax}_{a \in \mathcal{A}} \left\{ R(s, a) + \min_{s_\dagger \in \mathcal{S}_\dagger} \left\{ \hat{V}_\tau(s_\dagger, l_a, t+1) + L_{t+1} \|s_\dagger - s_a\| \right\} \right\}
$$

Then, Eq. 16 implies that

$$
|\hat{V}_\tau(s, l, t) - \hat{V}_\tau(s', l, t)| \leq \left| R(s, a^*) + \min_{s_\dagger \in \mathcal{S}_\dagger} \left\{ \hat{V}_\tau(s_\dagger, l_{a^*}, t+1) + L_{t+1} \|s_\dagger - s_{a^*}\| \right\} \right.
$$

$$
\left. - R(s', a^*) - \min_{s_\dagger \in \mathcal{S}_\dagger} \left\{ \hat{V}_\tau(s_\dagger, l_{a^*}, t+1) + L_{t+1} \|s_\dagger - s'_{a^*}\| \right\} \right|
$$

$$
\leq |R(s, a^*) - R(s', a^*)|
$$

$$
+ \left| \min_{s_\dagger \in \mathcal{S}_\dagger} \left\{ \hat{V}_\tau(s_\dagger, l_{a^*}, t+1) + L_{t+1} \|s_\dagger - s_{a^*}\| \right\} \right.
$$

$$
\left. - \min_{s_\dagger \in \mathcal{S}_\dagger} \left\{ \hat{V}_\tau(s_\dagger, l_{a^*}, t+1) + L_{t+1} \|s_\dagger - s'_{a^*}\| \right\} \right|
$$

$$
\quad (17)
$$

Now, let $\tilde{s}$ be the $s_\dagger \in \mathcal{S}_\dagger$ that minimizes the second part of the above subtraction, *i.e.*,

$$
\tilde{s} = \operatorname*{argmin}_{s_\dagger \in \mathcal{S}_\dagger} \left\{ \hat{V}_\tau(s_\dagger, l_{a^*}, t+1) + L_{t+1} \|s_\dagger - s'_{a^*}\| \right\}.
$$

As a consequence and in combination with Eq. 17, we get

$$
|\hat{V}_\tau(s, l, t) - \hat{V}_\tau(s', l, t)| \leq |R(s, a^*) - R(s', a^*)|
$$

$$
+ \left| \hat{V}_\tau(\tilde{s}, l_{a^*}, t+1) + L_{t+1} \|\tilde{s} - s_{a^*}\| \right.
$$

$$
\left. - \hat{V}_\tau(\tilde{s}, l_{a^*}, t+1) + L_{t+1} \|\tilde{s} - s'_{a^*}\| \right|
$$

$$
= |R(s, a^*) - R(s', a^*)| + L_{t+1} |\|\tilde{s} - s_{a^*}\| - \|\tilde{s} - s'_{a^*}\||
$$

$$
\overset{(*)}{\leq} C_{a^*} \|s - s'\| + L_{t+1} \|s_{a^*} - s'_{a^*}\|
$$

$$
\overset{(**)}{\leq} C \|s - s'\| + L_{t+1} K_{u_t} \|s - s'\| = L_t \|s - s'\|,
$$

where in $(*)$ we use the triangle inequality and the fact that the SCM $\mathcal{C}$ is Lipschitz-continuous, and $(**)$ follows from Lemma 5.

**Algorithm 2:** Graph search via $A^*$

---

**Input**: States $\mathcal{S}$, actions $\mathcal{A}$, observed action sequence $\{a_t\}_{t=0}^{t=T-1}$, horizon $T$, transition function $F_{\tau,t}^+$, reward function $R$, constraint $k$, initial state $\boldsymbol{s}_0$, heuristic function $\hat{V}_\tau$.

**Initialize**: NODE $v_0 \leftarrow \{\text{"tuple"} : (\boldsymbol{s}_0, 0, 0), \text{"rwd"} : 0, \text{"par"} : Null, \text{"act"} : Null\}$,
$\qquad$ STACK $action\_sequence \leftarrow [\,]$
$\qquad$ QUEUE $Q \leftarrow \{root\}$
$\qquad$ SET $explored \leftarrow \emptyset$

**while** *True* **do**
$\quad$ $v \leftarrow \operatorname{argmax}_{v' \in Q}\{v'.rwd + \hat{V}_\tau(v'.tuple)\}; Q \leftarrow Q \setminus v$ ; $\quad$ /* Next node to visit */
$\quad$ **if** $v.tuple = (*, *, T)$ **then**
$\quad\quad$ **while** $v.par \neq Null$ **do**
$\quad\quad\quad$ $action\_sequence.push(v.act)$ ; $\quad$ /* Retrieve final action sequence */
$\quad\quad\quad$ $v = v.par$
$\quad\quad$ **end**
$\quad\quad$ **return** $action\_sequence$
$\quad$ **end**
$\quad$ $explored \leftarrow explored \cup \{v\}$ ; $\qquad\qquad\qquad$ /* Set node $v$ as explored */
$\quad$ **if** $l = k$ **then**
$\quad\quad$ $availabe\_actions \leftarrow \{a_t\}$
$\quad$ **else**
$\quad\quad$ $availabe\_actions \leftarrow \mathcal{A}$
$\quad$ **end**
$\quad$ **for** $a \in available\_actions$ **do**
$\quad\quad$ $(\boldsymbol{s}, l, t) \leftarrow v.tuple$
$\quad\quad$ $(\boldsymbol{s}_a, l_a) \leftarrow F_{\tau,t}^+\left((\boldsymbol{s}, l), a\right)$ ; $\qquad\qquad$ /* Identify $v$'s children nodes */
$\quad\quad$ $v_a \leftarrow \{\text{"tuple"} : (\boldsymbol{s}_a, l_a, t+1), \text{"rwd"} : v.rwd + R(\boldsymbol{s}, a), \text{"par"} : v, \text{"act"} : a\}$
$\quad\quad$ **if** $v_a \notin Q$ *and* $v_a \notin explored$ **then**
$\quad\quad\quad$ $Q \leftarrow Q \cup \{v_a\}$ ; $\qquad$ /* Add them to the queue if unexplored */
$\quad\quad$ **end**
$\quad$ **end**
**end**

---

# E  A* algorithm

Algorithm 2 summarizes the step-by-step process followed by the $A^*$ algorithm. Therein, we represent each node $v$ by an object with $4$ attributes: (i) the "tuple" $(\boldsymbol{s}, l, t)$ of the node, (ii) the total reward "rwd" of the path that has led the search from the root node $v_0$ to the node $v$, (iii) the parent node "par" from which the search arrived to $v$, and (iv) the action "act" associated with the edge connecting the node $v$ with its parent. In addition to the queue of nodes to visit, the algorithm maintains a set of explored nodes, and it adds a new node to the queue only if it has not been previously explored. The algorithm terminates when the goal node is chosen to be visited, *i.e.*, the "tuple" attribute of $v$ has the format $(*, *, T)$, where $*$ denotes arbitrary values. Once the goal node $v_T$ has been visited, the algorithm reconstructs the action sequence that led from the root node $v_0$ to the goal node and returns it as output.

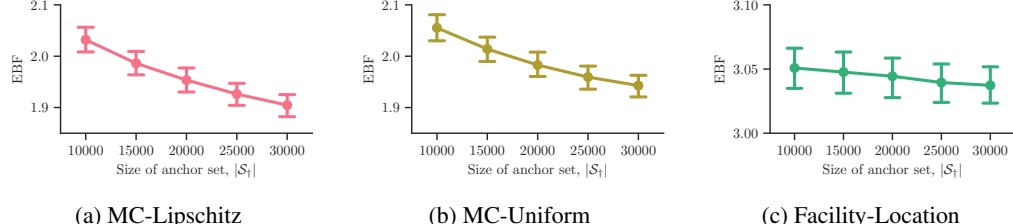

|         (a) MC-Lipschitz         |         (b) MC-Uniform         |         (c) Facility-Location         |

Figure 5: Computational efficiency of our method, measured by means of the Effective Branching Factor (EBF), under three different anchor set selection strategies against the size of the anchor set $\mathcal{S}_\dagger$. In all panels, we set $L_h = 1.0$, $L_\phi = 0.1$ and $k = 3$. Error bars indicate $95\%$ confidence intervals over 200 executions of the $A^*$ algorithm for 200 patients with horizon $T = 12$.

## F  Experimental evaluation of anchor set selection strategies

In this section, we benchmark the anchor set selection strategy presented in Section 4 against two alternative competitive strategies using the sepsis management dataset and the same experimental setup as in Section 5. More specifically, we consider the following anchor set selection strategies:

(i) *MC-Lipschitz*: This is the strategy described in depth in Section 4, based on Monte Carlo simulations of counterfactual episodes under randomly sampled counterfactual action sequences. Notably, the time steps where each counterfactual action sequence differs from the observed one are sampled proportionally to the respective Lipschitz constant $L_t$ of the SCM's transition mechanism. To ensure a fair comparison with other strategies, instead of controlling the number of sampled action sequences $M$, we fix the desired size of the anchor set $\mathcal{S}_\dagger$, and we repeatedly sample counterfactual action sequences until the specified size is met.

(ii) *MC-Uniform*: This strategy is a variant of the previous strategy where we sample the time steps where each counterfactual action sequence differs from the observed one uniformly at random, rather than biasing the sampling towards time steps with higher Lipschitz constants $L_t$.

(iii) *Facility-Location*: Under this strategy, the anchor set is the solution to a minimax facility location problem defined using the observed available data. Let $\mathcal{S}_o$ be the union of all state vectors observed in all episodes $\tau$ in a given dataset. Then, we choose an anchor set $\mathcal{S}_\dagger \subset \mathcal{S}_o$ of fixed size $|\mathcal{S}_\dagger| = b$, such that the maximum distance of any point in $\mathcal{S}_o$ to its closest point in $\mathcal{S}_\dagger$ is minimized. Here, the rationale is that counterfactual states resulting from counterfactual action sequences for one observed episode are likely to be close to the observed states of some other episode in the data. Formally,

$$\mathcal{S}_\dagger = \underset{\mathcal{S}' \subset \mathcal{S}_o : |\mathcal{S}'| = b}{\operatorname{argmin}} \left\{ \max_{\boldsymbol{s} \in \mathcal{S}_o} \min_{\boldsymbol{s}' \in \mathcal{S}'} \left\{ ||\boldsymbol{s} - \boldsymbol{s}'|| \right\} \right\}. \tag{18}$$

Although the above problem is known to be NP-Complete, we find a solution using the farthest-point clustering algorithm, which is known to have an approximation factor equal to 2 and runs in polynomial time. The algorithm starts by adding one point from $\mathcal{S}_o$ to $\mathcal{S}_\dagger$ at random. Then, it proceeds iteratively and, at each iteration, it adds to $\mathcal{S}_\dagger$ the point from $\mathcal{S}_o$ that is the furthest from all points already in $\mathcal{S}_\dagger$, *i.e.*, $\mathcal{S}_\dagger = \mathcal{S}_\dagger \cup \boldsymbol{s}$, where $\boldsymbol{s} = \max_{\boldsymbol{s}' \in \mathcal{S}_o} \left\{ \min_{\boldsymbol{s}_\dagger \in \mathcal{S}_\dagger} ||\boldsymbol{s}' - \boldsymbol{s}_\dagger|| \right\}$. The algorithm terminates after $b$ iterations.

**Results.** We compare the computational efficiency of our method under each of the above anchor set selection strategies for various values of the size of the anchor set $|\mathcal{S}_\dagger|$. Figure 5 summarizes the results. We observe that the Facility-Location selection strategy performs rather poorly compared to the other two strategies, achieving an effective branching factor (EBF) higher than 3. In contrast, the MC-Lipschitz and MC-Uniform strategies achieve an EBF close to 2, which decreases rapidly as the size of the anchor set increases. Among these two strategies, the MC-Lipschitz strategy, which we use in our experiments in Section 5, achieves the lowest EBF.

Table 2: Levels of vasopressors and intravenous fluids corresponding to the 25 actions in $\mathcal{A}$

| Vasopressors (mcg/kg/min) | Intravenous fluids (mL/4 hours) |
|:---:|:---:|
| 0.00 | 0 |
| 0.04 | 30 |
| 0.113 | 80 |
| 0.225 | 279 |
| 0.788 | 850 |

# G   Additional details on the experimental setup

## G.1   Features and actions in the sepsis management dataset

As mentioned in Section 5, our state space is $\mathcal{S} = \mathbb{R}^D$, where $D = 13$ is the number of features. We distinguish between three types of features: (i) demographic features, whose values remain constant across time, (ii) contextual features, for which we maintain their observed (and potentially varying) values throughout all counterfactual episodes and, (iii) time-varying features, whose counterfactual values are given by the SCM $\mathcal{C}$. The list of features is as follows:

- Gender (demographic)
- Re-admission (demographic)
- Age (demographic)
- Mechanical ventilation (contextual)
- FiO$_2$ (time-varying)
- PaO$_2$ (time-varying)
- Platelet count (time-varying)
- Bilirubin (time-varying)
- Glasgow Coma Scale (time-varying)
- Mean arterial blood pressure (time-varying)
- Creatinine (time-varying)
- Urine output (time-varying)
- SOFA score (time-varying)

To define our set of actions $\mathcal{A}$ we follow related work [6, 55, 57], and we consider 25 actions corresponding to $5 \times 5$ levels of administered vasopressors and intravenous fluids. Specifically, for both vasopressors and fluids, we find all non-zero values appearing in the data, and we divide them into 4 intervals based on the quartiles of the observed values. Then, we set the 5 levels to be the median values of the 4 intervals and 0. Table G.1 shows the resulting values of vasopressors and fluids.

## G.2   Additional details on the network architecture & training

We represent the location and scale functions $h$ and $\phi$ of the SCM $\mathcal{C}$ using neural networks with 1 hidden layer, 200 hidden units and $tanh$ activation functions. The mapping from a state $\boldsymbol{s}$ and an action $a$ to the hidden vector $\boldsymbol{z}$ takes the form $\boldsymbol{z} = tanh(W_s\boldsymbol{s} + W_a\boldsymbol{a})$, where $\boldsymbol{a}$ is a 2-D vector representation of the respective action. The mapping from the hidden vector $\boldsymbol{z}$ to the network's output is done via a fully connected layer with weights $W_z$. To enforce a network to have a Lipschitz constant $L$ with respect to the state input, we apply spectral normalization to the weight matrices $W_s$ and $W_z$, so that their spectral norms are $\|W_s\|_2 = \|W_z\|_2 = 1$. Additionally, we add 2 intermediate layers between the input and the hidden layer and between the hidden layer and the output layer, each one multiplying its respective input by a constant $\sqrt{L}$. Since it is known that the $tanh$ activation function has a Lipschitz constant of 1, it is easy to see that, by function composition, the resulting network is guaranteed to be Lipschitz continuous with respect to its state input with constant $L$. Note

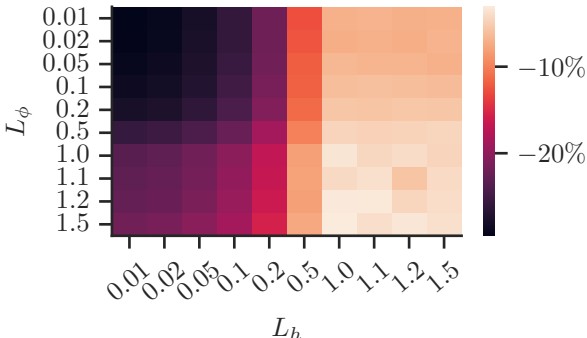

Figure 6: Goodness of fit of the Lipschitz-continuous SCM $\mathcal{C}$, measured by means of the percentage decrease in log likelihood of the data in comparison with an SCM trained without Lipschitz-continuity constraints. The x and y axes correspond to different enforced values for the Lipschitz constants $L_h$, $L_\phi$ of the location and scale networks $h$ and $\phi$, respectively. Darker values indicate that the learned SCM achieves a significantly lower log likelihood than the unconstrained SCM.

that, since the matrix $W_a$ is not normalized, the network's Lipschitz constant with respect to the action input can be arbitrary.

To train the SCM $\mathcal{C}$, for each sample, we compute the negative log-likelihood of the observed transition under the SCM's current parameter values (*i.e.*, network weight matrices & covariance matrix of the multivariate Gaussian prior), and we use that as a loss. Subsequently, we optimize those parameters using the Adam optimizer with a learning rate of $0.001$, a batch size of $256$, and we train the model for $100$ epochs.

We train the model under multiple values of the Lipschitz constants $L_h$, $L_\phi$ of the location and scale networks, and we evaluate the log-likelihood of the data under each model using 5-fold cross-validation. Specifically, for each configuration of $L_h$ and $L_\phi$, we randomly split the dataset into a training and a validation set (with a size ratio 4-to-1), we train the corresponding SCM using the training set, and we evaluate the log-likelihood of the validation set based on the trained SCM. This results in the log-likelihood always being measured on a different set of data points than the one used for training. For each configuration of $L_h$ and $L_\phi$, we repeat the aforementioned procedure 5 times and we report the average log-likelihood achieved on the validation set. In addition, we train an unconstrained model without spectral normalization, which can have an arbitrary Lipschitz constant.

Figure 6 shows the decrease in log-likelihood of the respective constrained model as a percentage of the log-likelihood achieved by the unconstrained model, under various values of the Lipschitz constants $L_h$, $L_\phi$. We observe that, the model's performance is predominantly affected by the Lipschitz constant of the location network $L_h$, and its effect is more pronounced when $L_h$ takes values smaller than 1. Additionally, we can see that the scale network's Lipschitz constant $L_\phi$ has a milder effect on performance, especially when $L_h$ is greater or equal than 1. Since we are interested in constraining the overall Lipschitz constant of the SCM $\mathcal{C}$, in our experiments in Section 5, we set $L_h = 1$ and $L_\phi = 0.1$, which achieves a log-likelihood only $6\%$ lower to that of the best model trained without any Lipschitz constraint.

