# OpenReview forum: "Finding Counterfactually Optimal Action Sequences in Continuous State Spaces"
_NeurIPS.cc/2023/Conference — NeurIPS 2023 poster_

### Official Review · Reviewer_rmB3 · 2023-07-03

**Soundness:** 2 fair
**Presentation:** 2 fair
**Contribution:** 3 good
**Rating:** 6
**Confidence:** 4

**Summary:**

The authors present a method for finding c/f (optimal) action sequences in sequential decision making problems with uncertainty, with the novelty that they consider continuous state dynamics. They apply their method to the interesting setting of sepsis treatment at the end of the paper and therein demonstrate impressive results for their method (compared to the actual, observed, action sequences).

**Strengths:**

Please see the Questions section for the full review.

**Weaknesses:**

Please see the Questions section for the full review.

**Questions:**

## Abstract

- I re-read the abstract multiple times and I unfortunately struggle with the first half. Consider changing the sentence construction or just simplify the first half. It is very unclear what you are trying to say and I fear that the positioning of the paper suffers because of it. One suggestion could be to anchor the paper with one or two real examples and then take the justification from there.
- The first paragraph of the introduction does a much better job of framing your paper, I would suggest adapting that approach for the first half of the abstract.

## Introduction

- Spell out ICU before you use it; it is common-enough term in the West but may be unfamiliar to a non-native speaker (or indeed anyone who does not frequently use English).
- Excellent second paragraph.
- Line 49: an SCM does not have the concept of state or transitions, merely a distribution P() over the endogenous and exogenous variables in the model (as defined originally by Pearl). Consequently how you have chosen to introduce it here does not quite follows. Consider re-phrasing. Moreover, SCMs are, as originally defined, state-less representations of a causal environment.
- It would be helpful to the reader and yourselves if you could show us (the readers) a quadrant of work which has considered MDPs with: (continuous states, discrete actions), (continuous states, continuous action) … and so on — so that we understand what gap you are filling with this work (certainly you spell it out, but I would argue that a quadrant would be more useful and visually more powerful).
- Line 60: please explain what a bijective SCM is. It is not enough to simply cite the original paper it if it is going to be key to your method.

##	A Causal Model of Sequential Decision Making Processes

- Line 76: presumably $a_t \in \mathcal{A} = \mathbb{Z}_+$?
- How come you are not considered the discounted cumulative reward?
- I think you should differentiate your setting more than what you are currently doing. At present it sounds as if eq (2) is part of the standard SCM definition. Perhaps introduce a definition paragraph for your SCM? That way the reader is clear that your are formalising a new concept and there is now ambiguity between yours and Pearl’s definition.
- I would like to see a much longer discussion on why you make the causal sufficiency assumption (Line 97) in this work (no unobserved confounders). It is a common assumption as you say, but merely saying that other works make the same assumption, is not good enough justification to make it a tenable assumption. It is a huge assumption and highly unrealistic in most realistic scenarions. It makes life computationally far easier but trades-off usability down the line. Consequently, please discuss why you make this assumption.
- Line 104: the do() express does not have to constitute a hard (atomic) intervention, it can also be a soft intervention (and others). Please consider or at least mention these settings too and why you settle for hard interventions.
- Definition 1: is this definition somehow different from the standard one? Perhaps you ought to also give it a source since this is common fare but it may differ from the one readers are used to (and would also elucidate on the matter if it the standard definition or not).
- The paragraph at the bottom of page 3 is great, very interesting application of Lipschitz continuity on a real problem.
- Define the indicator function in equation 5.
- Lines 149-161: I wonder if it may be not better for you to formally have a small lemma here, possibly even a theorem, explicating the results for which you have a proof. It currently reads as if it was merely a by-product of your method, whereas I imagine it is rather more important than that?

## Problem statement

- I would recommend revising the structure of your paper given that the problem statement appears on page four at conference with a page limit of nine. The paper will read far better if you frame the problem early on, allowing the reader to understand your angle of attack from the very start.
- Line 163: you already introduced the episode on line 78.
- Where is practically relevant to know that eq 9 is NP hard? A trivial question no doubt, but I would be interested to hear what the authors have in mind with regards to this theorem (is it really a theorem?)

## Finding the optimal counterfactual action sequence via A* search

- Line 210: I think a diagram here would be very helpful, explaining algorithm 1; a lot of page six could gainfully be explained with a figure rather than paragraphs of (rather dense) text.

## Experiments using clinical sepsis management data

- Looking at figure 2(a) I would have thought that as k increased we would have seen a much larger increase in the y-axis but this does not seem to be the case, the increase is fairly modest. Can you explain?
- According to panel 2(b), 50% of patients see less than or equal to ~5% c/f improvement? Is that correctly read?
- I enjoyed this section and the example is good but it is long and arduous, and we have no evidence to suggest that your method works for more (even synthetic) settings. It is an old trope of reviewing that we (reviewers) always want more experiments but in this case I think it is a fair request. Simply because I am struggling to understand how your ideas generalise across different settings and applications (even synthetic) nor do we have a comparison against other methods (are there any to which you can gainfully compare against?)


**Limitations:**

Please see the Questions section for the full review.

---

> ### Author Rebuttal · Authors · 2023-08-07
>
> We thank the reviewer for their careful and insightful comments, which will help improve our paper. Please, find a point-by-point response below.
>
> **[Miscellaneous comments on presentation]** We would like to thank the reviewer for all their concrete suggestions regarding presentation/organization of the paper. We will address all of them in the revised version of the paper.
>
> **[Line 49]** We would like to clarify that we do not refer to states or transitions as concepts inherent to the original SCM definition. As lines 48-49 mention, we *use* an SCM (as originally defined by Pearl) to *represent* the stochastic state transitions of a sequential decision making environment. For more details, please refer to our response to **[SCM definition]**.
>
> **[Bijective SCM]** Please note that we formally define what a bijective SCM is in Definition 2 (lines 136-138).
>
> **[Line 76]** $\mathcal{A}$ is a discrete *finite* set of actions, not the set of positive integers.
>
> **[Discounted cumulative reward]** Since we focus on a finite horizon setting, we think it is more meaningful to consider the undiscounted cumulative reward.
>
> **[SCM definition]** We would like to highlight that, in lines 82-84, where we first introduce the notion of an SCM, we do use Pearl’s (general) definition, i.e., that an SCM is consisted of four parts: (i) a set of endogenous variables (ii) a set of exogenous (noise) variables (iii) a set of structural equations assigning values to the endogenous variables, and (iv) a set of prior distributions characterizing the exogenous variables. Subsequently, in lines 84-92, we formally introduce a particular type of SCM $\mathcal{C}$ that we use to represent sequential decision making processes—an instance of Pearl’s general definition. In our SCM, each state $S_t$ and each action $A_t$ in a finite sequence of actions and states $\\{S_t, A_t\\}_ {t=0}^{T-1}$ corresponds to a different endogenous variable (see lines 84-86), the variables $\\{Z_t, U_t\\}_ {t=0}^{T-1}$ are exogenous variables characterized by prior distributions $\\{P^\mathcal{C} (Z_t), P^\mathcal{C} (U_t)\\}_ {t=0}^{T-1}$ (see lines 88-89 & 91-92), and the value of each variable $A_t$ ($S_t$) is given by a structural equation $g_A$ ($g_S$), as stated in Equation 1 (2). In the revised version of our paper, we will rephrase lines 82-84 to clarify that they refer to the general definition of an SCM given by Pearl.
>
> **[No unobserved confounders]** We agree with the reviewer that the assumption of no unobserved confounding may be more or less realistic, depending on the exact domain and application at hand. In our work, our focus is on developing a method to solve the problem of finding counterfactually optimal action sequences in continuous state spaces. Since this problem has not been studied before, for simplicity, we have decided to tackle the problem under the assumption that there are no unobserved confounders, similarly as others in the literature have done when addressing a problem for the first time. However, we do not think that assumption nullifies our contribution, and we hope that future research will build upon our work to develop methods that work under unobserved confounders. Following the reviewer’s advice, in the revised version of our paper, we will include a separate paragraph at the end of Section 2, where we will discuss the implications of this assumption and highlight related work that focuses on environments where unobserved confounding exists (currently discussed in Appendix A in the supplementary).
>
> **[Hard interventions]** We settle on hard interventions because they fit better the problem formulation we consider. In the revised version of the paper, we will clarify that it would be interesting to consider variations of our problem using soft interventions.
>
> **[Definition 1]** To the best of our knowledge, the specific definition of Lipschitz-continuity for the type of SCMs we consider has not been used before.
>
> **[NP-hardness]** It is practically relevant to know that Eq. 9 is NP-hard because it tells us that there is no hope in designing a polynomial time algorithm that solves the problem we focus on. Since the problem has not been proven to be NP-hard before and the reduction we use to prove NP-hardness is novel, we think it is reasonable to include such a result as a Theorem. In the revised version of our paper, after Theorem 3, we will include a paragraph where we will discuss the importance and implications of the Theorem in the design of our algorithm.
>
> **[Figure 2(a)]** Figure 2(a) shows that, as $k$ increases, the marginal gains that could have been achieved in terms of the total counterfactual reward are diminishing. That finding follows our intuition since it indicates that, in retrospect, a small number of actions in each episode had the most significant effect on the episode’s outcome.
>
> **[Figure 2(b)]** The reviewer reads panel 2(b) correctly, 50% of patients see less or equal to ~5% c/f improvement. As discussed in lines 348-349, this indicates that the treatment choices made by the clinicians for most of the patients were close to optimal, even with the benefit of hindsight.
>
> **[Other experimental settings, other methods]** As an implication of Theorem 6, our method is guaranteed to *always* find the optimal solution to the problem we study, and we do not have reasons to expect that it would perform poorly in a different dataset or application. That said, we would like to share that we did consider performing synthetic experiments prior to the submission of the paper, and we chose not to go forward because we believe there are no additional insights one could gain about our method in a synthetic environment, other than the ones already presented in the paper. Due to the rebuttal's space constraints, we kindly ask the reviewer to refer to the response **[Lack of baselines]** to Reviewer **R23o** for more details re. other methods (or lack thereof) to compare our method against.

---

> > ### Comment · Reviewer_rmB3 · 2023-08-15
> >
> > Thank you for your detailed response.
> >
> > I am happy to raise my score but will note that I do not think the authors have it right on the unobserved confounder assumption. Whilst I accept the logic for making this assumption here, in the first pass at the problem, it is and remains a significant assumption that rarely holds true save for toy problems. But I agree with the authors that this assumption, nonetheless, does not nullify your contribution.
> >
> > Again, well done, it is a fine paper.

---

> > > ### Author Response · Authors · 2023-08-16
> > > **Thank you for engaging in the discussion**
> > >
> > > We would like to thank the reviewer for engaging in the discussion and for updating their score. When revising the paper, we will make sure to highlight the assumption of no unobserved confounders and its implications, as mentioned in the rebuttal.

---

### Official Review · Reviewer_qYbP · 2023-07-06

**Soundness:** 3 good
**Presentation:** 4 excellent
**Contribution:** 3 good
**Rating:** 7
**Confidence:** 3

**Summary:**

This paper tackles the problem of finding counterfactual action sequences in sequential decision making problems. The main difference to previous methods is that this paper regards a continuous state space instead of a discrete one, which renders previous solution methods infeasible. The authors formalize the problem as an SCM and show that the solution in general is NP hard. As a solution, they present a new method based on the A-star algorithm, which yields for many problems an efficient solution. They evaluate the method on clinical data.

**Strengths:**

The paper is well-written and easy to follow.

The problem of efficiently finding counterfactual action sequences for the continuous state space seems new to me and relevant to the community.

Even if the SCM formulation follows closely the work of Tsirtsis et al. (2021), a likewise dynamic programming approach would require enumerating all possible action sequences. From my point of view, the main contribution lies therefore in the A-star-based search method (and the anchor set selection), which requires further technical details such as bijectivity and Lipschitz continuity of the SCM for the heuristics. Therefore, the technical contribution seems good to me.

The method is evaluated with regard to the influence of different parameters.

**Weaknesses:**

The worse case complexity of the method is the same as for brute-force search. The efficiency of the method depends on the number of samples of the Monte Carlo samples in the anchor set. It is not clear to me, which conditions the problem must have in order for the method to work effiently and when it falls back to worse case complexity.

Minor (typos):
line 190 comma before "such that"

**Questions:**

How important are the assumptions that the SCM is bijective and Lipschitz-continuous? You need this assumption for the bounds in your algorithm. Can you still approach the problem if one condition is violated? How much do you think do the conditions restrict the field of applications of your method?

Any thoughts about the conditions of the problem under which the method works efficiently?

**Limitations:**

Limitations and directions for future work are adequately addressed.

---

> ### Author Rebuttal · Authors · 2023-08-07
>
> We thank the reviewer for their careful and insightful comments, which will help improve our paper. Please, find a point-by-point response below.
>
> **[Efficiency of our method]** The efficiency of our method depends on the tightness of the bounds $\hat{V}_ {\tau}$, which depends (i) on the number of the Monte Carlo samples in the anchor set, and (ii) on the Lipschitz constant of the SCM. Therefore, in general, our method will work more efficiently the lower the Lipschitz constant of the SCM, as shown in Figure 1(a). However, one cannot just decrease the Lipschitz constant arbitrarily to increase efficiency because this would degrade the goodness of fit of the SCM with respect to observational data. In Figure 4 in Appendix F, we investigate the goodness of fit of the SCM under different values of the Lipschitz constants.
>
> **[Bijectivity and Lipschitz-continuity]** The assumptions that the SCM is bijective and Lipschitz continuous are both necessary to conclude that the solution returned by our algorithm is optimal and that our algorithm is efficient. Since the bijectivity assumption is satisfied by many classes of SCMs studied in the literature, as discussed in lines 132-135, and the Lipschitz continuity is a natural assumption, as discussed in lines 122-131, we do not think these two assumptions significantly restrict the field of applications of our method.

---

> > ### Comment · Reviewer_qYbP · 2023-08-20
> >
> > Thank you for answering my questions. I still think this is a good paper and therefore keep my original score for acceptance.

---

### Official Review · Reviewer_DTwm · 2023-07-07

**Soundness:** 1 poor
**Presentation:** 1 poor
**Contribution:** 2 fair
**Rating:** 6
**Confidence:** 2

**Summary:**

The paper tackles the problem of finding optimal action sequences in domains with continuous state spaces with counterfactual reasoning.  They propose an A* based search approach and show the efficacy of the approach on a clinical sepsis management problem.

**Strengths:**

* The paper focuses on a very significant research problem
* The experimental evaluation on a  real-world clinical dataset demonstrates the strong utility of the approach

**Weaknesses:**

* I have some concerns regarding the causal d-separation assumption (see Q1 & Q2).

**Questions:**

1. Graphical implication of an intervention on a variable $X$ (ie.  $do(X=x)$ ) is that all the incoming edges to $X$ are removed. In other words, as we are intervening on X, the influence from the $PA_X$  (parent nodes of X) on the node $X$ becomes zero. Refer to paragraph after Def 3.2.1 in Chapter 3 of Pearl. In Equation 3, it seems that all the arrows pointing out of $A_t$ are removed to adjust for $do(A_t = a_t)$ (as mentioned in line 106). So, the d-separation assumed does not follow from the do-calculus.
2. Similar d-separation assumptions are made in Eq. 4 and 6.
	1. Equation 1. suggests that two variables have a parent-child relationship: $S_t \rightarrow A_t$.
	2. Equation 2 suggests that their variable form a collider: $S_t \rightarrow S_{t+1} \leftarrow A_{t}$.
	3. On performing do-operation on $A_t$  would remove the parent-child relationship. But still maintain the collider  $S_t \rightarrow S_{t+1} \leftarrow A_{t}$. So, $S_{t+1}$ and $A_t$ are not d-separated by $S_t$.
3. What are the implications of modifying the d-separation assumption on the proposed algorithms?
4. How was the generated counterfactual action sequence evaluated? The SOFA score requires vital signs for calculations, how were the vital signs obtained?
5. Figure 1 needs a legend. It is not clear what the pink & green lines represent.
6. Did any physician analyze the counterfactual action sequences? Was any physian involved at any stage of the process?
7. I recommend authors provide a reference table for notations in the appendix, it was really difficult to keep track of all the notations used in the paper.

**Limitations:**

---

> ### Author Rebuttal · Authors · 2023-08-07
>
> We thank the reviewer for their careful and insightful comments, which will help improve our paper. Please, find a point-by-point response below.
>
> **[d-separation]** Although the reviewer is correct that, in general, a do() operation on a variable (in our case, $A_t$) removes all *incoming* edges to that variable (in our case, $\mathbf{S}_ t \rightarrow A_t$), we respectfully disagree that Eq. 3 does not follow from the do-calculus. On the contrary, the first equality of Eq. 3 is a direct application of the 2nd rule of the do-calculus (action/observation exchange), as given in Theorem 3.4.1 of Pearl [8]. Specifically, the rule (applied to our case) states that one can write the (observational) probability $p^\cal{C}(\mathbf{S}_ {t+1} = \mathbf{s} | \mathbf{S}_ t = \mathbf{s}_ t, A_t = a_t)$ as an interventional probability $p^{\cal{C} ; do(A_t=a_t)} (\mathbf{S}_ {t+1} = \mathbf{s} | \mathbf{S}_ t = \mathbf{s}_ t)$ and vice versa if $\mathbf{S}_ {t+1}$ is conditionally independent of $A_t$ given $\mathbf{S}_ t$ in the graph resulting after deleting all *outgoing* edges of $A_t$. Note that this is equivalent to what lines 104-106 of our submission state—in the graph $A_t \leftarrow \mathbf{S}_ t \rightarrow \mathbf{S}_ {t+1}$, where the original outgoing edge $A_t \rightarrow \mathbf{S}_ {t+1}$ is deleted, $\mathbf{S}_ t$ acts as a confounder and conditioning on it d-separates $A_t$ and $\mathbf{S}_ {t+1}$.
>
> In the revised version of our paper, we will include a figure illustrating the causal graph, and we will rephrase lines 104-106 to clarify this point. In light of the aforementioned explanation, we do not see any technical issues with Eqs. 3, 5 and 6, and we would be grateful if the reviewer could reconsider their score.
>
> **[Evaluation based on SOFA score]** As discussed in lines 344-346, the generated counterfactual action sequences were evaluated based on the counterfactual improvement they would have provided according to the SCM $\mathcal{C}$, i.e., the relative decrease in cumulative SOFA score between the counterfactual and the observed episode. The SOFA score and the eight vitals required to compute it form the dimensions of the state vector, as stated in lines 305-307, and their counterfactual values are given by the (trained) SCM.
>
> **[Figure 1]** To avoid occlusions, we specify what the pink and green lines represent in the caption of Figure 1. The pink line represents the effective branching factor (EBF) and the green line represents the  average runtime of the A* search (in seconds). Here, we also used matching colors in the left and right y-axis of each panel to further indicate that the pink line corresponds to EBF and the green line corresponds to A* average running time.
>
> **[Evaluation by physicians]** The counterfactual action sequences were not analyzed by physicians for the purposes of this submission. In our work, we focus on formalizing and tackling algorithmically the problem of finding counterfactually optimal action sequences for episodes of sequential decision making processes with continuous states. However, as discussed in lines of 370-373, performing a user study with a systematic evaluation of counterfactual action sequences by human experts is a very interesting direction for future work.
>
> **[Notation]** Following the reviewer’s advice, in the revised version of our paper, we will include a reference table for notations in the Appendix.

---

> > ### Comment · Reviewer_DTwm · 2023-08-11
> > **D-separation**
> >
> > Thank you for the clarification on the D-separation assumption. That was my only major concern. As this is addressed, I will update my score accordingly.

---

> > > ### Author Response · Authors · 2023-08-15
> > > **Thank you for engaging in the discussion**
> > >
> > > We would like to thank the reviewer for engaging in the discussion and for updating their score.

---

### Official Review · Reviewer_MQT2 · 2023-07-25

**Soundness:** 3 good
**Presentation:** 3 good
**Contribution:** 2 fair
**Rating:** 5
**Confidence:** 3

**Summary:**

This paper studies a question that is very natural for a sequential decision maker to ask itself: "how could I best improve the return I obtained in a trajectory by only changing a fixed (k) number of actions from the sequence of actions I just executed while keeping the rest of the actions fixed". Authors study this question in a context where the state space is continuous but the action space is discrete. The paper generalizes the result of previous work which studied a similar question, but in the context of discrete states. The setting is very nicely motivated and the problem is quite relevant and important.

This formulation seems to be novel, general, and useful. However, the paper shows that answering this question is in fact an NP-hard, one that we can only solve in the worst case using exhaustive search. Therefore, the paper goes on to find an admissible heuristic to be used in conjunction with the A* algorithm to reduce the complexity of the search space.

To propose a consistent/admissible heuristic, one needs to find reasonable upper bounds for the value of unseen states which are found using counterfactual reasonings. The paper proposes to get this done using the notion of Lipschitz continuity where intuitively, the difference between the value of the two states is upper bounded by a constant times the norm of the difference of the two states. This may be a stringent requirement generally, but it seems to make sense in the domain that is of interest in the paper, namely in medical trials. Experiments show that using this heuristic does result in significantly reducing the average runtime of the search.

Overall, this is a fair contribution in the intersection of causal reasoning and planning/reinforcement learning.

**Strengths:**

The competency that the paper attempts to develop, namely being able to answer counterfactual questions in the context of planning and reinforcement learning, is quite interesting. Formulating this problem and showing that the original problem is NP-hard also sheds some light on the complexity of asking such questions. Some of the assumptions made, such as using a bijective causal model does limit the scope of the result, but I still find the formulation very interesting.



**Weaknesses:**

We know that Lipchitz continuity can be a pretty lose upper bound for the function value. Especially in the context of this paper, when data is sparse, the upper bounds derived in Lemma 4 can be pretty bad. Moreover, recently there have been more efforts to develop Lipschitz-like tools that are more conducive to RL algorithms by ensuring that the new tool are Coarse-Grained. See for example "Coarse-Grained Smoothness for Reinforcement Learning in Metric Spaces", by Gottesman et al, 2023 or "Zooming for Efficient Model-Free Reinforcement Learning in Metric Spaces" by Touati et al 2020. This paper though relies on more classical notions of smoothness (vanilla Lipchitz continuity) which is not the most effective in the context of RL.


Another point is that under the assumptions made in this paper (namely Lipschitz reward and transition dynamics), the value function itself becomes Lipschitz, so when learning the value function one can restrict the set of feasible value functions to be Lipschitz also. Is there any reason why this is not leveraged? In the same vein, we clearly know that the value function is always between (1-\gamma) R_max and -(1-\gamma) R_max. Can you elaborate if the bounds that are obtained by the Lipschitz interpolation would be able to provide much tighter bounds or that the bounds become vacuous? Can you also elaborate on the amount of data needed in your medical domains before these bound become non vacuous?

In practice, we also need to make sure that we do not underestimate the Lipschitz constant, because otherwise the heuristic used in A* will no longer be admissible. The estimate also needs to be tight enough so as to make sure that it is effective when used as a heuristic. But I am not sure how this trade-off will be maintained in problems where the Lipschitz constant is unknown.


**Questions:**

I would like to see a comparison with the discrete case, in particular the "Woulda, coulda, shoulda: Counterfactually-guided policy search" paper of Buesing et al, 2018. I do understand that this paper is dealing with the continuous state space. But one natural baseline to compare against could for example be the paper mentioned above where one can simply discretize the continuous state space and then apply their algorithm to compare against.

How important is the kick-starting part with the monte carlo anchoring? In particular, it would have been nice to see ablations where the monte-carlo anchoring is performed more or less to understand its effect on performance.

**Limitations:**

Authors maintain that they will release code for their experiments if accepted, but they seem to have access to confidential patient data. I hope and trust that authors will they take sufficient precautions to ensure the anonymity of patients whose information is used in these experiments.

---

> ### Author Rebuttal · Authors · 2023-08-07
>
> We thank the reviewer for their careful and insightful comments, which will help improve our paper. Please, find a point-by-point response below.
>
> **[Lipschitz-like tools for RL algorithms]** We would like to thank the reviewer for bringing these papers to our attention, which we will cite in the revised version of our paper. However, we would like to highlight that, in our paper, we do not learn a model-free RL policy nor study the relationship between smoothness, Lipschitz continuity and the amount of data needed to learn a model-free RL policy. Rather, given an observed sequence of actions and states, we search for an alternative sequence of actions that would have retrospectively maximized the (counterfactual) outcome under the counterfactual transition dynamics given by a (trained) structural causal model (SCM)---this is stated verbally in lines 40-42 and formalized in lines 180-183. In other words, our paper studies an *algorithmic* problem and not a *learning* problem. Therefore, it is hard to make a direct comparison with the papers brought up by the reviewer. That said, we would also like to clarify that the bounds obtained in Lemma 4 do *not* depend on the sparsity of the available data and they are an implication of the Lipschitz continuity of the SCM (see Definition 1).
>
> **[Lipschitz value function]** In our paper, to guarantee that the computed heuristic function $\hat{V}_ \tau$ is provably consistent, we do leverage the assumption that the SCM is Lipschitz continuous (see Lemma 4, Proposition 5, Theorem 6 and the related proofs in Appendix C). Also, we would like to clarify that our approach does not involve *learning* a value function. Note that the function $V_ \tau$ does not denote the value function, as typically defined in the RL literature, but denotes the counterfactual reward that could have been achieved in a counterfactual episode where, at time $t$, the process is at a (counterfactual) state $s$, and there are so far $l$ actions that have been different in comparison with the observed action sequence in the observed episode $\tau$ (see lines 192-195).
>
> To our best understanding, the upper bound $(1- \gamma ) R_{max}$ mentioned by the reviewer is not an upper bound relevant to our function $V_\tau$ but rather a bound for the value function in an RL problem with an infinite horizon. Note that, in our work, we consider a finite horizon (see lines 31, 60 & 75), and the outcome of the decision making process is the sum of the rewards $o(\tau)=\sum_t R(\mathbf{s}_t, a_t)$, which implies that $\gamma=1$. Therefore, together with the explanation given in the previous paragraph, we believe that the bound given by the reviewer is not applicable to our problem.
>
> Finally, we would also like to clarify that, to compute a better heuristic function $\hat{V}_\tau$, we do not require more data to get better bounds, but rather more anchor points, which we get via Monte Carlo simulations (see lines 277-295).
>
> **[Underestimation of Lipschitz constant]** In our experiments, we do not estimate the Lipschitz constant but rather we train an SCM that is Lipschitz-continuous by design and whose Lipschitz constant we can control. For a detailed description of this process and the overall model architecture, please refer to Appendix F2 in the supplementary.
>
> **[Comparison with "Woulda, coulda, shoulda…"]** We would like to clarify that “Woulda, coulda, shoulda: Counterfactually-guided policy search” by Buesing et al. [10] solves a different problem and thus it is incomparable with our method. In this context, a natural baseline to compare against would be the method introduced by Tsirtsis et al., NeurIPS 2021 [13] which solves a closely related problem in discrete state spaces, as we discuss in lines 52-54. Unfortunately, their method has a quadratic complexity with respect to the number of discrete states and thus does not scale to continuous multidimensional vector states as those used in our experiments. For example, discretizing the 9 continuous features we consider in our experiments into ten discrete levels each---a rather coarse-grained discretization---would lead to 1 billion discrete states.
>
> **[Monte Carlo anchoring]** The kick-starting approach we describe in lines 277-295 is an important part of our method because it generates the anchor set $\mathcal{S}_ \dagger$ required to compute the heuristic function $\hat{V}_ \tau$. Note that, in Figure 1(b), we vary the amount of Monte Carlo anchoring to investigate its effect on performance and, in Appendix E in the supplementary, we evaluate alternative selection strategies for the anchor set.
>
> **[Patient data]** We will release the code we used in our experiments but not the *anonymized* patient dataset (i.e., the MIMIC-III dataset). In this context, we would like to clarify that we are not the data owners, and we have gained access to the MIMIC-III dataset by submitting an application to the official owners (Physionet). As part of the process, we have passed a short online class in ethics. Please, refer to the MIMIC-III website for more details.

---

> > ### Comment · Reviewer_MQT2 · 2023-08-17
> > **Lipschitz Nets**
> >
> > Thanks for the pointer to the Appendix. What I originally meant is that the Lipschitz constants $L_{\phi}$ and $L_h$ could be estimated based on the data. The discussion on how these are chosen is currently pretty limited. For example, the Appendix reads that the Lipschitz models are only 6% worse in terms of log-likelihood. What does this mean? Is this log-likelihood on a single set of data, or log-likelihood on a held out test data? Moreover, when applying this approach to a new domain, how do I know what is a reasonable $L_{\phi}$ and $L_h$?

---

> > > ### Author Response · Authors · 2023-08-18
> > > **Response to follow-up question**
> > >
> > > We would like to thank the reviewer for engaging in the discussion. Perhaps this is already clear to the reviewer, but we would like to highlight that our goal is not to use data to train and estimate the Lipschitz constant of a *single* SCM, since that could lead to an underestimation of its value and could end up being problematic for the optimality of our proposed method, as correctly mentioned by the reviewer in the original review. Instead, our approach is to train *multiple* SCMs that are Lipschitz continuous by design—each one is *guaranteed* to consist of neural networks $h$ and $\phi$ with Lipschitz constants $L_h$ and $L_\phi$ whose values we can control. Then, we evaluate the log-likelihood achieved by each of these SCMs using 5-fold cross validation, as discussed in lines 831-832, and pick the SCM with the best tradeoff between log-likelihood, $L_h$ and $L_\phi$. This procedure is described in lines 830-841 in Appendix F in the supplementary.
> > >
> > > To evaluate the log-likelihood using 5-fold cross validation, for each configuration of $L_h$ and $L_\phi$, we randomly split the dataset into a training and a validation set (with a size ratio 4-to-1), we train the corresponding SCM using the training set, and we evaluate the log-likelihood of the validation set based on the trained SCM. We repeat the procedure 5 times and we take the average of the achieved log-likelihoods. In the revised version of our paper, we will clarify that the log-likelihood is always measured on a different set of data points than the one used for training.
> > >
> > > Whenever we write that the Lipschitz models are only 6% worse in terms of log-likelihood, we mean that the log-likelihood achieved by the Lipschitz models is only 6% lower than the log-likelihood achieved by a baseline model whose Lipschitz constants are unconstrained.
> > >
> > > To pick $L_h$ and $L_\phi$ in a new domain, we would repeat the aforementioned procedure using data from the new domain and defer to domain experts to elucidate what is an acceptable trade-off between log-likelihood, $L_h$ and $L_\phi$.

---

### Official Review · Reviewer_R23o · 2023-07-26

**Soundness:** 3 good
**Presentation:** 4 excellent
**Contribution:** 3 good
**Rating:** 7
**Confidence:** 3

**Summary:**

The paper studies the problem of finding a counterfactual action sequence to maximize the outcome of a trajectory in an MDP characterized by a structural causal model. The focus is on continuous state spaces under a set of Lipschitz constraints. The authors show that this problem is NP hard and propose an algorithm based on A* search, which they test empirical on a dataset on clinical decision making.

**Strengths:**

- The paper studies a relevant well-motivated problem that will be of interest to the NeurIPS community
- The authors do a good job in clarifying and formalizing the problem
- Showing that the proposed problem is NP-hard is a valuable insight to guide future theoretical and empirical work
- The proposed algorithm is a natural choice and it's formulation is sound
- The empirical evaluation is performed on a realistic dataset, and the results are promising

**Weaknesses:**

- The discussion of related work is too limited. I think there is a lot of related work about related problems in SCM (outside RL) that should be mentioned. I'm not very familiar with this literature myself, which made it difficult to judge the potential impact of the contribution the present paper makes.

- Theorem 3 (NP-hardness) is a key contribution of the paper. However, it is only stated in the paper and almost not discussed. It would be valuable to give a high level picture of the reduction used to prove the result, as well as a discussion of the implications of this result and how it results in the choice of algorithm later in the paper.

- The empirical results are difficult to interpret due to a lack of baselines. While the efficiency results in Figure 1 and performance results in Figure 2 look as expected qualitatively, there is no point of comparison.
   - For efficiency it would be useful to compare to some naive baselines, like a search based method without a specific heuristic.
   - The counterfactual performance improvements in Figure 2 are pretty small which makes me wonder if the dataset is a good enough benchmark to evaluate the present method. Maybe the trajectories in the dataset are too close to optimal?
   - It could be interesting to compare to an approach the discretizes the environment and uses prior methods for discrete state spaces.

**Questions:**

- What could be baselines to compare the empirical results to?
   - What impact does the chosen heuristic have on the performance of the search algorithm?
   - How much worse would be a method that discretizes the environment?

- How can we be confident that the present dataset is a good benchmark to evaluate the approach?

**Limitations:**

Yes

---

> ### Author Rebuttal · Authors · 2023-08-07
>
> We thank the reviewer for their careful and insightful comments, which will help improve our paper. Please, find a point-by-point response below.
>
> **[Related work]** We agree with the reviewer that there is a rich literature on structural causal models (SCMs) that is not connected to reinforcement learning (RL). Due to space limitations, we chose to focus the discussion only on the most closely related work [10-13] in the main body of our paper and include a further discussion of related work in Appendix A. In the revised version of our paper, we will bring some of the related work from Appendix A to the main body, and we will add a footnote explicitly pointing the reader to Pearl [8] for a more complete overview of prior work on causal inference based on SCMs.
>
> **[NP-Hardness]** The main implication of the NP-Hardness result is that there is no hope in designing a polynomial time algorithm that solves the problem. Following the reviewer’s advice, after Theorem 3, we will include a paragraph where we will give a high level picture of the reduction used in the proof of the Theorem and discuss the importance and implications of the Theorem in the design of our algorithm.
>
> **[Lack of baselines]** We thank the reviewer for their concrete experimental suggestions. To the best of our knowledge, the only natural baseline to compare against would be the method introduced by Tsirtsis et al., NeurIPS 2021 [13] which solves a closely related problem in discrete state spaces, as we discuss in lines 52-54. Unfortunately, their method has a quadratic complexity with respect to the number of discrete states and thus does not scale to continuous multidimensional vector states as those used in our experiments. For example, discretizing the 9 continuous features we consider in our experiments into ten discrete levels each---a rather coarse-grained discretization---would lead to 1 billion discrete states.
>
> In terms of naive baselines, we believe it is non-trivial to come up with methods other than an exhaustive search that simultaneously work without a heuristic function and guarantee finding the optimal solution. We do not directly compare our method with an exhaustive search because it is too computationally expensive to run in a reasonable amount of time. That said, our experimental results indicate that our method does perform better than an exhaustive search since, otherwise, it would be exploring the entire search space independently of the exact form of our computed heuristic function. For example, varying the number of Monte Carlo samples $M$ would have no effect on efficiency, i.e., Figure 1(b) would be a flat line.
>
> **[Quality of the sepsis management dataset]** We decided to perform experiments using the MIMIC-III sepsis management dataset since it is a commonly used dataset in the literature on reinforcement learning for healthcare (see lines 297-298) and it is relevant to our problem motivation. Note that, as an implication of Theorem 6, our method is guaranteed to *always* find the optimal solution, and we do not have reasons to expect that it would perform poorly using a different dataset. However, we would like to clarify that the reviewer’s comment that *``maybe the trajectories in the dataset are too close to optimal''* is part of our experimental findings (see lines 347-349) and we do not view this as an indicator that the dataset is not a good benchmark for the evaluation of our method.

---

> > ### Comment · Reviewer_R23o · 2023-08-12
> >
> > Thanks for the response! The responses address my primary concerns, and if the author update the paper as promised, I think the paper would be improved quite a bit. I appreciate the comments regarding the difficulty to choose suitable baselines; I think this should also be discussed in the paper.
> >
> > Overall, I am still a bit skeptical about how reliable the experimental results are; especially, because it is only a single dataset. But given that my other concerns were addressed, I will increase my score from 6 to 7 (assuming the authors make the promised changes to the paper).

---

> > > ### Author Response · Authors · 2023-08-15
> > > **Thank you for engaging in the discussion**
> > >
> > > We would like to thank the reviewer for engaging in the discussion and for updating their score. We will make sure to perform all edits mentioned in the rebuttal, when revising our paper.

---

### Decision · Program_Chairs · 2023-09-21

**Decision:**

Accept (poster)

**Comment:**

The reviewers unanimously agreed on acceptance.